# An exported protein-interacting complex involved in the trafficking of virulence determinants in *Plasmodium*-infected erythrocytes

Steven Batinovic[1], Emma McHugh[1,*], Scott A. Chisholm[2,*], Kathryn Matthews[2], Boiyin Liu[1], Laure Dumont[1], Sarah C. Charnaud[3], Molly Parkyn Schneider[1], Paul R. Gilson[3], Tania F. de Koning-Ward[2], Matthew W.A. Dixon[1,**] & Leann Tilley[1,**]

The malaria parasite, *Plasmodium falciparum*, displays the *P. falciparum* erythrocyte membrane protein 1 (*Pf*EMP1) on the surface of infected red blood cells (RBCs). We here examine the physical organization of *Pf*EMP1 trafficking intermediates in infected RBCs and determine interacting partners using an epitope-tagged minimal construct (*Pf*EMP1B). We show that parasitophorous vacuole (PV)-located *Pf*EMP1B interacts with components of the PTEX (*Plasmodium* Translocon of EXported proteins) as well as a novel protein complex, EPIC (Exported Protein-Interacting Complex). Within the RBC cytoplasm *Pf*EMP1B interacts with components of the Maurer's clefts and the RBC chaperonin complex. We define the EPIC interactome and, using an inducible knockdown approach, show that depletion of one of its components, the parasitophorous vacuolar protein-1 (PV1), results in altered knob morphology, reduced cell rigidity and decreased binding to CD36. Accordingly, we show that deletion of the *Plasmodium berghei* homologue of PV1 is associated with attenuation of parasite virulence *in vivo*.

[1] Department of Biochemistry and Molecular Biology, Bio21 Institute, The University of Melbourne, Parkville, Victoria 3010, Australia. [2] School of Medicine, Deakin University, Waurn Ponds, Victoria 3220, Australia. [3] Macfarlane Burnet Institute for Medical Research and Public Health, Melbourne, Victoria 3004, Australia. * These authors contributed equally to this work. ** These authors jointly supervised this work. Correspondence and requests for materials should be addressed to L.T. (email: ltilley@unimelb.edu.au).

Plasmodium falciparum causes more than 200 million cases of malaria each year and kills ∼438,000 people[1]. The deaths are due to malaria complications that particularly affect young children and pregnant women. *P. falciparum* develops inside the red blood cells (RBCs) of its human host, imparting an adhesive phenotype on infected RBCs, leading to receptor binding on the microvascular endothelium[2]. Adhesion sequesters the infected RBCs from the circulation, thereby preventing the clearance of infected RBCs in the spleen. An exacerbated inflammatory response to the sequestered parasites can precipitate complications that lead to coma and death[3].

Adhesion of *P. falciparum*-infected RBCs is mediated by a parasite-derived multiprotein complex containing the major virulence protein, the *P. falciparum* erythrocyte membrane protein-1 (*Pf*EMP1)[4,5]. *Pf*EMP1 is anchored into knob-like structures via its C-terminal cytoplasmic domain, the acidic terminal segment (ATS). The extracellular domains of *Pf*EMP1 contain the variable DBL (Duffy binding-like) and CIDR (cysteine-rich inter-domain region) domains that facilitate binding to host receptors[6]. *Pf*EMP1 is encoded by the epigenetically controlled 60-member *var* multigene family[7]. Switching expression between different *Pf*EMP1 variants allows the parasite to change its surface-exposed antigen, thus evading the host's protective antibody response[8]. Therefore, *Pf*EMP1 is central to both disease pathology and acquisition of immunity; however, little is known about the export pathways used to traffic *Pf*EMP1 to the RBC surface.

To reach the host RBC surface, *Pf*EMP1 must be exported beyond the confines of the parasite itself; crossing the parasite plasma membrane (PPM), the parasitophorous vacuole membrane (PVM) and finally transiting through the RBC cytoplasm. In the parasitophorous vacuole (PV) it is hypothesized that *Pf*EMP1 undergoes translocation across the PVM via the PTEX (*Plasmodium* Translocon of EXported proteins)[9]. Indeed, conditional knockdown of the PTEX components heat shock protein-101 (HSP101) or PTEX-150 (PTEX150) prevents *Pf*EMP1 export beyond the PVM[10,11]. However, it remains to be determined whether *Pf*EMP1 is a direct substrate for PTEX or is dependent on other proteins that are exported via PTEX[12].

Two previous studies of the *Pf*EMP1 trafficking process have made use of transfectants expressing genome-integrated green fluorescent protein (GFP)-tagged minimal *Pf*EMP1 constructs[13,14]. We have used one of these constructs (here referred to as *Pf*EMP1B). In transfectants expressing *Pf*EMP1B, a proportion of the chimeric protein is delivered to the RBC surface[13], while some remains associated with intermediate trafficking compartments, including the PV and the Maurer's clefts—a pattern reminiscent of the distribution of endogenous *Pf*EMP1 (refs 14–17).

Here, we have used immunoprecipitation (IP) and mass spectrometry to describe the *Pf*EMP1B interactome in different cellular compartments. We identified a number of novel components of the *Pf*EMP1 trafficking pathway, including a complex of proteins located in the PV. We show that genetic

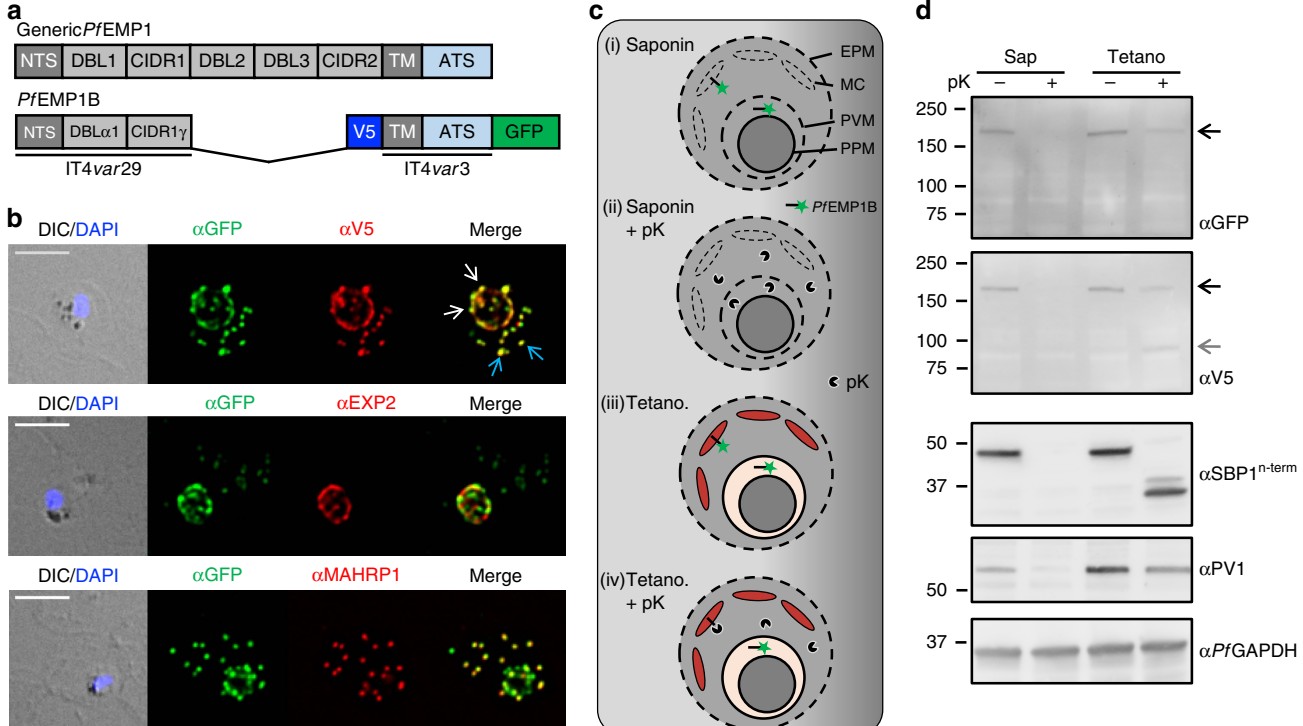

**Figure 1 | Location and protease accessibility of *Pf*EMP1B during export.** (**a**) Schematic of the minimal *Pf*EMP1 construct, *Pf*EMP1B, used in this study. *Pf*EMP1B contains the semiconserved head structure from IT4var29, an internal V5 epitope tag, a transmembrane (TM) domain and acidic terminal segment (ATS) from IT4var3 followed by GFP. Structure of a generic *Pf*EMP1 molecule is shown above. (**b**) Immunofluorescence analysis of *Pf*EMP1B-infected RBCs labelled with antisera recognizing GFP (green) and V5 or EXP2 or MAHRP1 (red). Differential interference contrast (DIC) image and parasite nuclei (stained with DAPI; blue) are shown on the left. White arrows indicate PV-localized *Pf*EMP1B while blue arrows indicate *Pf*EMP1B located at the Maurer's clefts. Scale bars, 5 μm. (**c**) Schematic of the protease protection assay performed on *Pf*EMP1B-infected RBCs. Cells were first treated with either saponin, to permeabilize the RBC membrane, the Maurer's clefts and PVM or with Tetanolysin (Tetano) to permeabilize only the RBC membrane, and then incubated in the presence or absence of proteinase K (pK). (**d**) Western blots of the protease-treated infected RBCs were probed with anti-GFP and anti-V5 antisera. SBP1, PV1 and *Pf*GAPDH western blots are shown as controls. Full-length *Pf*EMP1B is indicated with a black arrow and the truncated product with a grey arrow. Molecular masses are shown in kDa. Uncropped western blots are shown in Supplementary Fig. 12.

knockdown of the expression of one these components, parasitophorous vacuolar protein-1 (PV1), compromises host cell remodelling, decreases adhesion of *P. falciparum*-infected RBCs and renders *Plasmodium berghei* less virulent.

## Results

**A *Pf*EMP1 chimera accumulates in the PV and Maurer's clefts**. *Pf*EMP1B contains a semiconserved N-terminal head domain (NTS, DBL and CIDR segments), a transmembrane (TM) domain and a conserved acidic terminal segment domain (Fig. 1a), enabling successful delivery to the RBC surface[13]. The GFP and V5 epitope tags on *Pf*EMP1B are recognized in transgenic parasites expressing the chimera in a compartment with a 'necklace of beads' profile at the parasite periphery (Fig. 1b; white arrows), as well as in punctate structures in the RBC cytoplasm (Fig. 1b; blue arrows), in agreement with a previous report[13]. These compartments respectively represent the PV (where there is partial co-localization with PTEX component exported protein-2 (EXP2)) and the Maurer's clefts (co-localization with Maurer's cleft protein membrane-associated histidine-rich protein-1 (MAHRP1)) (Fig. 1b). Dual immunofluorescence labelling with HSP101, ring-exported protein-1 (REX1) and membrane-associated histidine-rich protein-2 (MAHRP2) also showed patterns consistent with those observed for endogenous *Pf*EMP1 (Supplementary Fig. 1a)[14,17–19].

**PV-located *Pf*EMP1B is not membrane embedded**. To investigate the physical organization of *Pf*EMP1B in the different compartments we performed a protease protection assay[16,20]. Samples were treated with either saponin, known to permeabilize the RBC, Maurer's cleft and PV membranes[21] or Tetanolysin (Tetano) that permeabilizes the RBC membrane but leaves both the Maurer's clefts and PVM intact[22], and incubated with or without proteinase K (schematic in Fig. 1c). In the absence of proteinase K, full-length *Pf*EMP1B is recognized as a band of ~175 kDa by antisera recognizing the GFP and V5 epitope tags (located on opposite sides of the TM domain) (Fig. 1d, black arrows). Upon treatment with proteinase K, most *Pf*EMP1B is digested leaving a small amount of the full-length protein (Fig. 1d). This remnant full-length *Pf*EMP1B represents a population that is fully protected (that is, in the parasite or PV). In addition, a truncated fragment was detected at a size of 90–95 kDa using V5 antisera after treatment with Tetanolysin (Fig. 1d, grey arrow). This fragment is consistent with protection of the N-terminal domain of *Pf*EMP1 embedded within the Maurer's clefts[16]. Upon treatment with saponin, which is known to permeabilize the Maurer's clefts and PVM but leave the PPM intact[21], the remaining full-length *Pf*EMP1B is degraded by proteinase K (Fig. 1d). This indicates the presence of a population of nonmembrane embedded *Pf*EMP1B in the PV. The effectiveness of the Tetanolysin and saponin treatments were tested by probing membranes with antisera raised against Maurer's cleft protein skeleton-binding protein-1 (SBP1), PV1 and *Pf*GAPDH (Fig. 1d, lower panels)[23–25]. Cleavage of SBP1 following Tetanolysin and saponin treatment is consistent with previous studies[26]. Cleavage of PV1 following saponin but not Tetanolysin treatment is consistent with integrity of the PVM following Tetanolysin treatment, while protection of *Pf*GAPDH demonstrates the integrity of the PPM in both treatments.

We additionally performed an immunofluorescence experiment to determine the accessibility of epitopes to antibodies following saponin treatment of cells in suspension[27] (Supplementary Fig. 1b). PV-located *Pf*EMP1B was labelled with GFP or V5 antisera following saponin treatment (Supplementary Fig. 1c).

**PfEMP1B interacts with components of the PTEX translocon**. We questioned whether the PV-located population of *Pf*EMP1B was in the process of export via PTEX. We first performed a proximity ligation assay (PLA) to determine whether *Pf*EMP1B is located close to PTEX. *Pf*EMP1B transfectant-infected RBCs that were co-labelled with anti-GFP and anti-HSP101 antisera produced a PLA signal at the parasite periphery (Fig. 2a), demonstrating close proximity of *Pf*EMP1B and HSP101 in the PV. Interestingly, a *Pf*EMP1B signal was also observed in regions distinct from the region of the PLA signal, indicating the majority of *Pf*EMP1B may reside in a separate compartment in the PV. Conversely, the control reaction using GFP antisera alone displayed *Pf*EMP1B labelling but no PLA signal (Fig. 2a).

To investigate whether *Pf*EMP1B interacts directly with PTEX components we performed IP using GFP-Trap. To control for nonspecific interactions we used a nonsecreted GFP-tagged minimal *Pf*EMP1 (here termed *Pf*EMP1F)[13] and an unrelated PV-located GFP-reporter (EXP1[1-35]-GFP) (Fig. 2b)[28]. IP of the different GFP fusion proteins was demonstrated by western blot (Supplementary Fig. 2a). A specific interaction between *Pf*EMP1B and PTEX was supported by the pull-down of HSP101 and EXP2 with *Pf*EMP1B, but not with *Pf*EMP1F and EXP1[1-35]-GFP controls (Fig. 2c).

**PfEMP1B interacts with SBP1 and REX1 at the Maurer's clefts**. Given that *Pf*EMP1 transits via the Maurer's clefts *en route* to the infected RBC surface, we again made use of *Pf*EMP1B co-immunoprecipitation to identify *Pf*EMP1 interactions occurring in these structures. We employed increasing concentrations of a membrane-permeable reversible amine-reactive crosslinker, DSP (dithiobis[succinimidyl propionate]), to enrich weaker protein interactions during IP. Pull-down of both SBP1 and REX1 was observed with *Pf*EMP1B but not with the *Pf*EMP1F chimera (Fig. 2d). The level of REX1 pull-down was found to increase (>10-fold) in the presence of DSP, suggesting a weaker or indirect interaction with *Pf*EMP1B. The specificity of these reactions was confirmed by performing the reverse experiment using antibodies raised against SBP1 and REX1 to pull-down crosslinked *Pf*EMP1B (Fig. 2e). PLA analysis reveals close proximity of these proteins in the Maurer's clefts (Supplementary Fig. 2b), indicating that this is the location of these interactions.

**Host and parasite proteins that interact with *Pf*EMP1B**. We used mass spectrometry to investigate global *Pf*EMP1B interactions. A portion of protein eluates from *Pf*EMP1B and *Pf*EMP1F pull-downs were resolved by SDS–polyacrylamide gel electrophoresis (SDS-PAGE) and silver stained. Numerous protein bands were visualized in both *Pf*EMP1F and *Pf*EMP1B samples including full-length *Pf*EMP1F and *Pf*EMP1B migrating at their respective molecular masses of 52 and 175 kDa (Fig. 3a; black arrows). An enrichment of protein bands was observed in the *Pf*EMP1B sample (Fig. 3a; examples marked with asterisks). Tandem mass spectrometry (MS/MS) analysis of the remaining eluates identified 66 parasite and human proteins that preferentially interact with *Pf*EMP1B (two independent experiments; Supplementary Tables 1 and 2).

We categorized peptides based on the localization or function of parent proteins and found *Pf*EMP1B pulled down proteins from all known compartments in its export pathway (Fig. 3b). All five validated components of PTEX (PTEX150, HSP101, EXP2, PTEX-88 (PTEX88) and thioredoxin-2 (TRX2)) were detected[9]. We normalized the MS/MS spectral counts representing peptides from PTEX proteins to the size of the respective proteins as a semiquantitative measure of the interaction affinity with *Pf*EMP1B and found HSP101 to be the most strongly associated

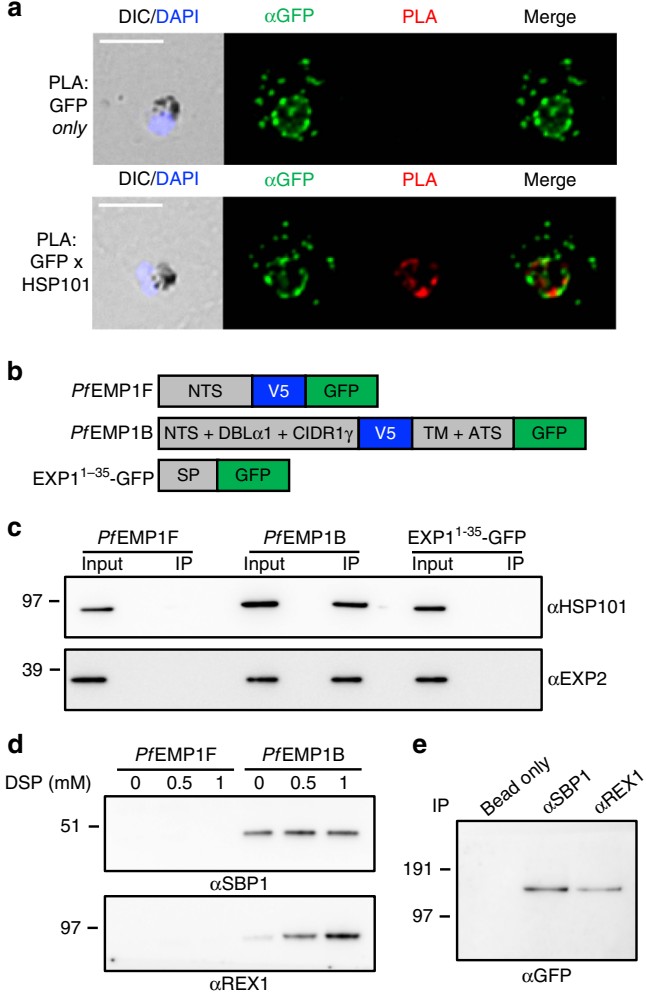

**Figure 2 | *Pf*EMP1B interacts with components of PTEX and the Maurer's clefts.** (**a**) Proximity ligation analysis (PLA) of *Pf*EMP1B-infected RBCs labelled with antisera recognizing GFP alone (top) or GFP and HSP101 (bottom). A PLA signal is only produced if the primary epitopes are ≤ 40 nm apart. DIC and DAPI (blue), anti-GFP (green) and PLA (red) are shown. Scale bars, 5 μm. (**b**) Schematic of GFP-fusion proteins used in immunoprecipitation (IP) experiments. *Pf*EMP1F contains the N-terminal sequence (NTS) from IT*var3* followed by a V5 epitope tag and GFP and is located in the parasite cytoplasm as previously described[16]. *Pf*EMP1B is as described in Fig. 1a. EXP1$^{1-35}$-GFP contains the signal peptide (SP) from EXP1 followed by GFP and localizes to the PV as previously described[47]. (**c**) Immunoprecipitation of *Pf*EMP1F, *Pf*EMP1B and EXP1$^{1-35}$-GFP transfectants using GFP-Trap. Total protein (input) and eluate (IP) were prepared for western blot and probed with anti-HSP101 and anti-EXP2 antisera. (**d**) Immunoprecipitation of *Pf*EMP1F and *Pf*EMP1B parasite lysates crosslinked using increasing concentrations of DSP. Western blots were probed with anti-SBP1 and anti-REX1 antisera. (**e**) Reverse IP on crosslinked *Pf*EMP1B parasite lysate using SBP1 and REX1 antisera. Western blots were probed with anti-GFP antisera. Precipitation of agarose beads with no antiserum was used as a control. (**c–e**) Molecular masses are shown in kDa. Uncropped western blots are shown in Supplementary Fig. 12.

PTEX component (Fig. 3c). Eleven exported parasite proteins (Supplementary Table 1) were detected, including SBP1 and REX1 in good agreement with our western blot and PLA data (Fig. 2d,e and Supplementary Fig. 2b) and other proteins such as MAHRP1 and *Pf*EMP1-trafficking protein-5 (PTP5), consistent with known roles of these proteins in *Pf*EMP1 export[29,30]. We provide further evidence for an interaction between endogenous

*Pf*EMP1 and MAHRP1, MAHRP2 and PTP5 by IP of GFP-tagged parasite lines (Supplementary Fig. 3a–c). We also performed dual labelling of PTP5-GFP transfectants using anti-GFP and anti-SBP1 antisera to demonstrate that PTP5 is located at the Maurer's clefts (Supplementary Fig. 3d).

A number of human proteins were also pulled down with *Pf*EMP1B but not *Pf*EMP1F (Fig. 3d and Supplementary Table 2). Eight components of the human chaperonin complex, the TCP-1 ring complex (TRiC), a known interaction partner of TRiC, HSP90 and a regulatory subunit of the 26S proteasome were found to be the most enriched human proteins (Fig. 3e)[31]. TRiC is a eukaryotic ATPase system responsible for the folding of multiple substrate cargo proteins[32–34], and may interact in a promiscuous fashion with unfolded exported proteins.

**Characterization of *P. falciparum* PV proteins.** Given that *Pf*EMP1B was found to accumulate in the PV, we used this construct as a tool to decipher additional export events in this compartment. In our mass spectrometric analysis, we found that in addition to PTEX, *Pf*EMP1B pulled down another PV-localized protein, PV1 (Supplementary Table 1)[25]. Immunofluorescence and western blot analysis confirmed the close association of PV1 and *Pf*EMP1B (Supplementary Fig. 3e,f). To functionally assess the interaction between *Pf*EMP1 and PV1, we generated a haemagglutinin (HA)-tagged inducible knockdown transgenic parasite line of PV1 using the *glmS* riboswitch system (Fig. 4a and Supplementary Fig. 4a)[35]. We similarly produced inducible knockdown lines for two additional *Pf*EMP1B-interacting proteins, PF3D7_1226900 and PF3D7_1024800. We refer to these proteins as parasitophorous vacuolar protein-2 (PV2) and exported protein-3 (EXP3), respectively (Fig. 4a). These proteins have predicted secretory signal sequences but no *Plasmodium* export element (PEXEL) sequence (Fig. 4a). Integration of the PV1, PV2 and EXP3 *glmS* vectors into the endogenous loci was confirmed by PCR analysis (Supplementary Fig. 4b) and the expression of these HA-fusion proteins assessed by western blot. Protein bands consistent with the predicted sizes of the mature PV1-HA (52 kDa), PV2-HA (59 kDa) and EXP3-HA (173 kDa) were observed (Fig. 4b). Immunofluorescence microscopy using anti-HA antisera confirmed that PV1-HA, PV2-HA and EXP3-HA are directed to the PV and co-localize with PV1 (Fig. 4c). Solubility analysis indicated that PV1 and PV2 were largely soluble in sodium carbonate buffer while EXP3 was only soluble in Triton X-100, consistent with its predicted transmembrane domain (Supplementary Fig. 5).

**A new protein complex located in the PV of *P. falciparum*.** Given the co-localization and solubility profiles of PV1, PV2 and EXP3, we performed blue native (BN)-PAGE analysis to investigate whether these proteins formed a protein complex. A dominant band migrating between 1,048 and 1,236 kDa was observed in all three transfectant lines (Fig. 5a, black arrow), suggesting that the proteins participate in a large protein complex. We refer to this complex as the EPIC (Exported Protein-Interacting Complex). PV1 and PV2 were also found to be present in smaller molecular mass complexes (see full-length gel in Supplementary Fig. 6a). We occasionally observed an anti-HA-labelled complex higher (> 1,236 kDa) than the dominant EPIC complex in PV1-HA and EXP3-HA samples (Fig. 5a, grey arrow), reminiscent of the previously described native PTEX complex[36,37]. To determine the relationship between EPIC and PTEX complexes, EXP3-HA BN-PAGE samples were probed with anti-HA and anti-HSP101 antisera. A small population of EXP3-HA was detected in the PTEX complex as recognized by anti-HSP101 antisera (Fig. 5b, grey arrow), while the majority was present in

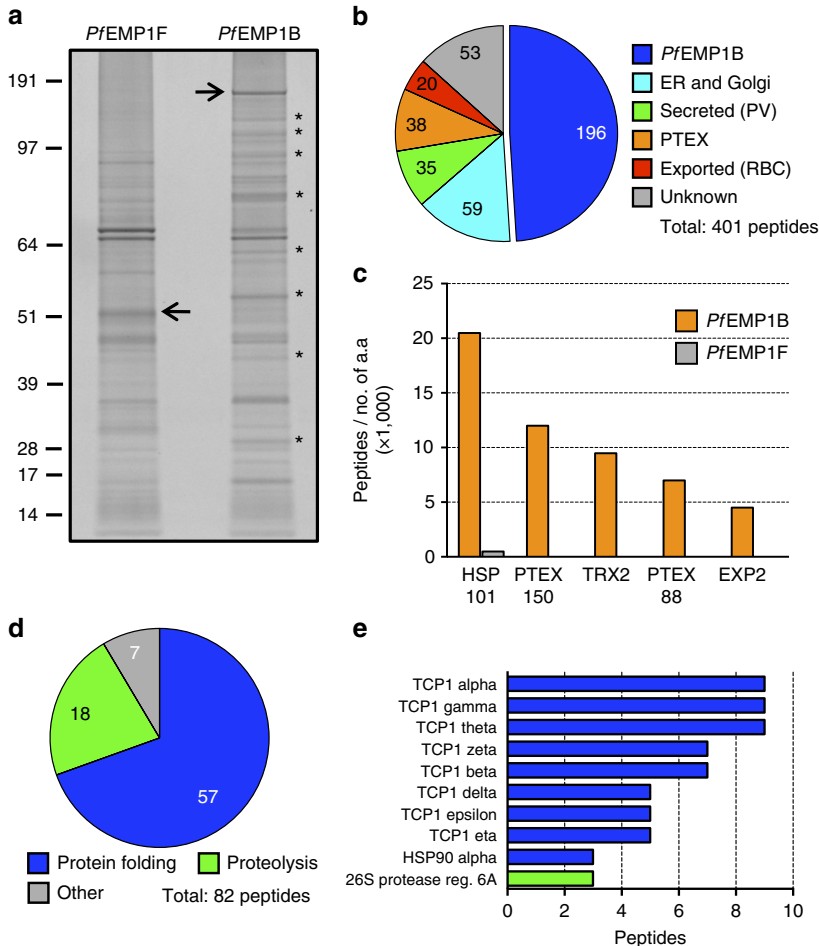

**Figure 3 | Global analysis of *Pf*EMP1B-interacting proteins.** (**a**) Silver-stained SDS-PAGE of material eluted from IPs of crosslinked *Pf*EMP1F and *Pf*EMP1B parasite lysates. Examples of protein bands enriched in the *Pf*EMP1B eluate are marked with asterisks. Full-length *Pf*EMP1F and *Pf*EMP1B are indicated with arrows. Molecular masses are shown in kDa. (**b**) Peptides identified by tandem mass spectrometry in *Pf*EMP1B IPs sorted according to function or known localization of the parent proteins. (**c**) Relative abundance of co-precipitated PTEX components with *Pf*EMP1B as determined by dividing the number of peptides identified by the number of amino acids (a.a.) of the individual PTEX proteins. (**d**) Human peptide matches identified in *Pf*EMP1B IPs sorted according to the functions of the parent proteins. (**e**) The top 10 human proteins (by fold enrichment) identified in *Pf*EMP1B IPs. (**b–e**) Peptide numbers are based on averages from two independent experiments. All identified proteins are listed in Supplementary Tables 1 and 2.

EPIC (Fig. 5b, black arrow), suggesting that components of EPIC may shuttle between complexes.

**Components of EPIC co-localize and co-precipitate with PTEX.** We further examined the interaction of EPIC proteins with PTEX by performing immunofluorescence microscopy. PV1, PV2 and EXP3 each show a high level of co-localization with HSP101 and EXP2 in the PV (Fig. 5c and Supplementary Fig. 6b). PV1-HA was additionally found to co-localize with HSP101 in merozoites (Supplementary Fig. 6c), suggesting that like PTEX components, PV1 is synthesized in schizonts and delivered to the PV during or shortly after invasion[36]. We next performed IP experiments with PV1-HA, PV2-HA and EXP3-HA transfectants using anti-HA-conjugated agarose beads. Both PV2-HA and EXP3-HA pulled down PV1 (Fig. 5d–f), in agreement with the immunofluorescence and BN-PAGE analysis. For each EPIC protein, an interaction with the PTEX components, HSP101 and EXP2, was demonstrated in transfectant lines but not in the wild type (Fig. 5d–f). Furthermore, *Pf*113, a recently described PTEX-associated protein[37], was pulled down with PV1-HA. The interaction of PV1 with PTEX proteins is in good agreement with a previous report[37]. In contrast, *Pf*GAPDH was not present in any of the eluates (Fig. 5d–f).

**Global mass spectrometric analysis reveals EPIC interactome.** To dissect the interactome of the EPIC, we performed mass spectrometric analysis on IP eluates from the three HA-tagged EPIC components. We identified 66 parasite proteins that preferentially interact with EPIC components (two independent experiments; Supplementary Tables 3–5). Proteins that were pulled down by all three EPIC components included PTEX components PTEX150, HSP101, PTEX88 and EXP2, merozoite surface protein-9 and exported proteins PIESP2 and FIKK 10.1 (Fig. 6a–c). The identification of these and other exported proteins in the EPIC pull-downs suggests a possible role for EPIC in preparing exported proteins for export through PTEX (Supplementary Tables 3–5). Proteins known to reside in the PV, including merozoite surface proteins and Ser-repeat antigens were also pulled down with one or more EPIC components. Endoplasmic reticulum (ER) and exported Hsp70s (Hsp70-2 and Hsp70-x) were found to be significantly enriched in PV1-HA eluates compared with the cytoplasmic Hsp70 (Hsp70-1), suggesting an interplay between EPIC and these chaperones (Fig. 6d,e).

**Knockdown of EPIC proteins does not affect parasite viability.** To examine the role of EPIC we made use of the *glmS* riboswitch

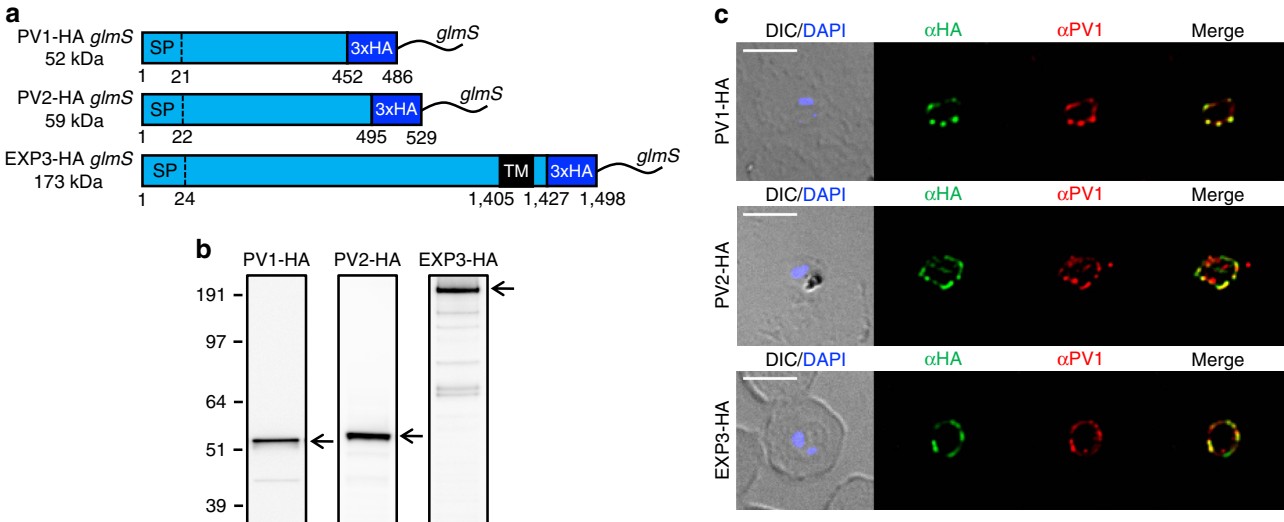

**Figure 4 | Generating HA-tagged knockdown lines of novel PV proteins.** (**a**) Schematic of endogenous PV1, PV2 and EXP3 proteins tagged with HA and *glmS*. Sizes of fusion proteins are indicated. All proteins contain predicted signal peptides (SPs) and EXP3 contains a predicted transmembrane (TM) domain. Numbers refer to amino acids. (**b**) Western blots were probed with anti-HA antisera to confirm PV1-HA, PV2-HA and EXP3-HA fusion protein expression in transfectants. Expected full-length fusion proteins are marked by arrows. Molecular masses are shown in kDa. (**c**) Immunofluorescence analysis of PV1-HA-, PV2-HA- and EXP3-HA-infected RBCs labelled with antisera recognizing HA (green) and PV1 (red). DIC and DAPI (blue) are shown on the left. Scale bars, 5 μm.

present in the PV1-HA, PV2-HA and EXP3-HA *glmS* transfectants (Fig. 4a). In the presence of the cofactor glucosamine-6-phosphate, the *glmS* ribozyme degrades mRNA encoding the targeted gene, reducing its expression[35]. We treated synchronous trophozoite-stage (26–34 h post invasion (h.p.i.)) transfectants with glucosamine (0 to 5 mM) and observed a dose-dependent knockdown of PV1-HA, PV2-HA and EXP3-HA in the subsequent cycle (Fig. 7a and Supplementary Fig. 7a,b; 86–96% apparent reduction in the signal at 2.5 mM). To determine whether EPIC components are required for growth *in vitro*, synchronous trophozoites (30–32 h.p.i.) were treated with glucosamine (0 mM or 2.5 mM) continuously for two consecutive asexual parasite cycles. No difference in growth was observed relative to the parental lines, indicating that EPIC components are either not essential or only needed in low amounts for normal growth *in vitro* (Supplementary Fig. 8).

**PV1 knockdown disrupts infected RBC remodelling.** To assess effects on the display of adhesins on the infected RBC surface we panned PV1-HA, PV2-HA and EXP3-HA *glmS* transfectants on immobilized CD36 to enrich the population for CD36-binding variants, and then examined their ability to adhere to immobilized CD36 under physiological flow conditions. Knockdown of PV1-HA *glmS* parasites with 2.5 mM glucosamine decreased the binding to CD36 compared with untreated parasites ($73 \pm 6\%$ s.e.m., $n = 5$, $P = 0.002$ according to Student's *t*-test; Fig. 7b), indicating that physiological levels of PV1 are needed for efficient adhesion to endothelial ligands. Glucosamine treatment had no effect on the 3D7 parent. No significant changes in adhesion to CD36 were observed for the PV2-HA and EXP3-HA *glmS* transfectant lines upon treatment with 2.5 mM glucosamine compared with the parental A4 strain (Supplementary Fig. 9a,b).

To further assess the virulence phenotype of PV1-HA *glmS* parasites after knockdown, we probed their physical deformability by subjecting infected RBCs (23–26 h.p.i.) to filtration through a bed of microbeads[38]. Knockdown of PV1-HA *glmS* parasites with 2.5 mM glucosamine led to a marked increase in cellular filterability ($48 \pm 6\%$ flow-through) compared with untreated parasites ($26 \pm 4\%$ flow-through; $n = 6$, $P = 0.01$ according to Student's *t*-test; Fig. 7c). In contrast, treatment of the parent line with 2.5 mM glucosamine had no effect on filterability (Fig. 7c).

To determine the molecular basis for the decreased cytoadherence and cellular rigidity, we used immunofluorescence signals as a semiquantitative measure of the level of protein export in the PV1 knockdown. Treatment of PV1-HA *glmS* with 2.5 mM glucosamine led to a significant decrease in the normalized fluorescence intensity of *Pf*EMP1 ($63\% \pm 8\%$ of nontreated control; $n \geq 15$; $P = 0.006$ according to Student's *t*-test), KAHRP ($67\% \pm 4.8\%$ of nontreated control; $n \geq 56$; $P = 0.001$ according to Student's *t*-test) and *Pf*EMP3 ($80\% \pm 6\%$ of nontreated control; $n \geq 26$; $P = 0.009$ according to Student's *t*-test) (Fig. 7d–f; Supplementary Fig. 9c–e). We next investigated knob morphology by scanning electron microscopy. PV1-HA *glmS* parasites treated with 2.5 mM glucosamine showed an increase in knob size ($108 \pm 2$ nm) compared with untreated ($95 \pm 2$ nm) ($n = 6$ cells, $>430$ measurements, $P = 0.003$ according to Student's *t*-test) (Fig. 7g,h), while no significant difference in knob size was observed upon treatment of the 3D7 parent (Fig. 7g,h). Taken together, these data indicate that PV1 is needed for efficient export of proteins that play important roles in remodelling the host RBC membrane skeleton.

**Loss of virulence upon genetic disruption of *P. berghei* PV1.** The *P. berghei* rodent model has been validated as a suitable tool to analyse factors involved in malaria virulence[11,39]. The *P. berghei* homologues of PV1 (PBANKA_0919100), PV2 (PBANKA_1441700) and EXP3 (PBANKA_0509000) were targeted for deletion. The targeting constructs (Supplementary Fig. 10a) were designed to integrate into the endogenous *P. berghei* loci by double crossover homologous recombination such that successful replacement of the targeted gene would result in expression of fluorescent mCherry protein under the endogenous promoter. All three loci were successfully targeted as confirmed by PCR (Supplementary Fig. 10b) and live cell imaging (Supplementary Fig. 10c).

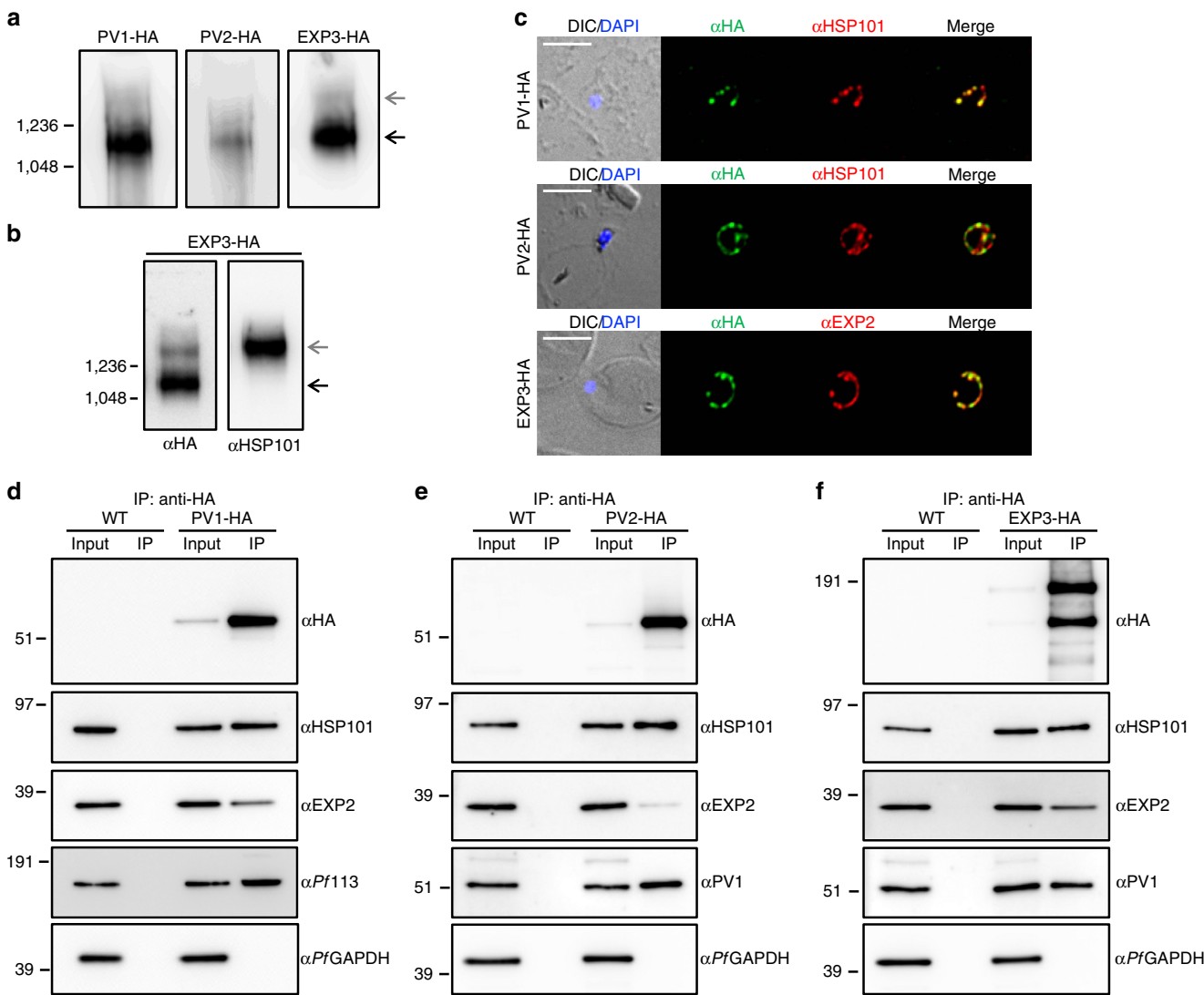

**Figure 5 | Identification of a novel PV protein complex that interacts with PTEX.** (**a,b**) Blue native (BN)-PAGE analysis of PV1-HA-, PV2-HA- and EXP3-HA-infected RBCs. Western blots were probed with (**a,b**) anti-HA and (**b**) anti-HSP101 antisera. Black and grey arrows indicate different size protein complexes. (**c**) Immunofluorescence analysis of PV1-HA-, PV2-HA- and EXP3-HA-infected RBCs labelled with antisera recognizing HA (green) and HSP101 or EXP2 (red). DIC and DAPI (blue) are shown on the left. Scale bars, 5 µm. (**d–f**) Immunoprecipitation of PV1-HA, PV2-HA and EXP3-HA parasite lysates using anti-HA-conjugated agarose beads. Total protein (input) and eluate (IP) were prepared for western blot and probed with (**d–f**) anti-HA, anti-HSP101, anti-EXP2, (**d**) anti-*Pf*113 and (**e,f**) anti-PV1 antisera. Membranes were probed with anti-*Pf*GAPDH antisera as a negative control. (**a,b,d–f**) Molecular masses are shown in kDa. Uncropped western blots are shown in Supplementary Fig. 12.

Two experiments were performed whereby C57BL/6 mice ($n = 6$ and $n = 11$ per group) were infected with $1 \times 10^6$ wild-type *Pb*ANKA or *Pb*ΔPV1, *Pb*ΔPV2 or *Pb*ΔEXP3 parasites. Tail blood was analysed daily from 3 days post infection by flow cytometry, revealing the parasitaemia of the mice infected with the knockout parasites remained similar to those infected with wild-type parasites as determined by an unpaired *t*-test (Fig. 7i; Supplementary Fig. 11a–c). Thus *Pb*PV1, *Pb*PV2 and *Pb*EXP3 are not essential for parasite replication *in vivo*. The development of cerebral malaria symptoms was also monitored in these mice. Two separate clones of mice infected with *Pb*ΔPV1 parasites succumbed to cerebral malaria 2–4 days later than mice infected with wild-type *Pb*ANKA (*Pb*ANKA 5–7 days post infection, *Pb*ΔPV1 7–9 days post infection) (Fig. 7j and Supplementary Fig. 11c). Statistical analysis of parasite survival was performed using a log-rank test (clone 1, $n = 11$, clone 2, $n = 6$ for *Pb*ANKA; clone 1 $n = 10$, clone 2, $n = 6$ for *Pb*ΔPV1), indicating a significant difference between treatment groups ($P < 0.0001$ and

$P = 0.0006$), suggesting that *Pb*PV1 contributes to parasite virulence. No difference in survival was observed in mice infected with *Pb*ΔPV2 and *Pb*ΔEXP3 parasites (Supplementary Fig. 11b).

## Discussion

Despite the importance of *Pf*EMP1 in both the pathology of malaria and as a target for protective immunity, the mechanism and route for trafficking of this adhesin to the RBC surface is poorly understood. We have made use of a *Pf*EMP1 minimal construct (*Pf*EMP1B) that we have previously shown accumulates in two compartments *en route* to the RBC surface, namely in a 'necklace of beads' compartment at the parasite periphery and at the Maurer's clefts in the RBC cytoplasm[14], as is the case for endogenous *Pf*EMP1 (refs 16,17). We found that the PV-localized *Pf*EMP1B population is nonmembrane embedded. In agreement with a previous report that showed that *Pf*EMP1 is carbonate extractable during transit through the parasite secretory system[40], our data indicate *Pf*EMP1 is only inserted into the membrane

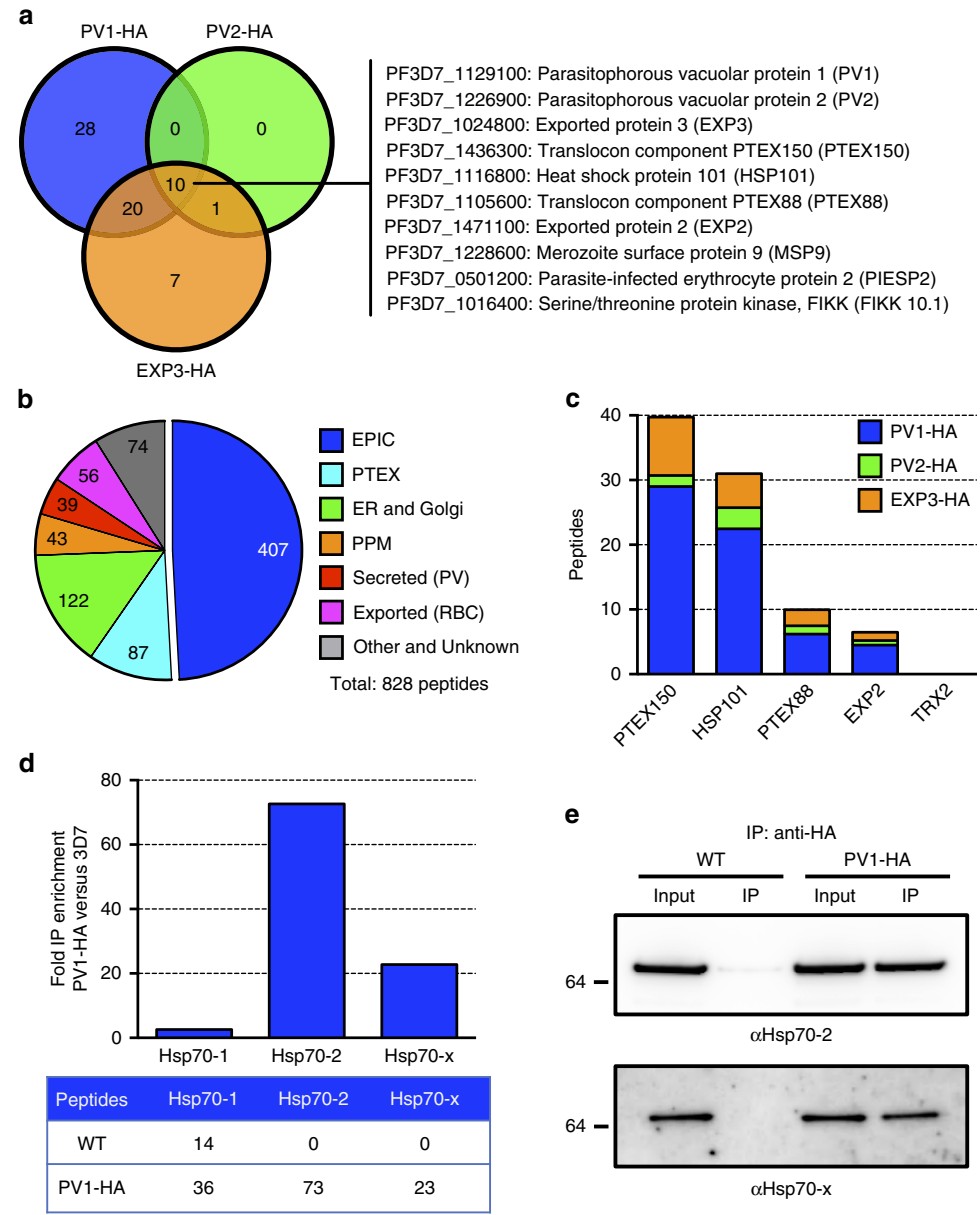

**Figure 6 | Analysis of the EPIC interactome.** (**a**) Venn diagram of proteins identified by tandem mass spectrometry of IPs of EPIC components (PV1-HA, PV2-HA and EXP3-HA) IPs. PlasmoDB accession numbers and names of proteins that were found to pull-down with all three bait proteins are shown. Numbers in the diagram refer to number of proteins pulled down with each bait protein. (**b**) Peptides identified by tandem mass spectrometry in EPIC component IPs sorted according to function or known localization of the parent proteins. (**c**) PTEX peptides pulled down with EPIC components. (**a–c**) Peptide numbers are based on averages from two independent experiments. All identified proteins are listed in Supplementary Tables 3–5. (**d,e**) Analysis of enrichment of *Plasmodium* HSP70 proteins in PV1-HA IPs. (**d**) Fold enrichment of Hsp70-1, Hsp70-2 and Hsp70-x peptides in PV1-HA IPs compared with wild type is shown. The table below shows average numbers of peptides identified from two independent experiments. (**e**) Immunoprecipitation of PV1-HA parasite lysate using anti-HA-conjugated agarose beads. Western blots of total protein (input) and IP eluate (IP) were probed with anti-HSP70-2 and anti-HSP70-x antisera. Molecular masses are shown in kDa. Uncropped western blots are shown in Supplementary Fig. 12.

when it reaches the Maurer's clefts. This is in contrast to other PEXEL-negative exported proteins (PNEPs), which are incorporated into the ER membrane and require extraction from the PPM to permit export across the PVM[22,41]. *Pf*EMP1 has a much larger molecular mass than most PNEPs that may preclude membrane insertion and extraction.

Given the extensive repertoire of exported proteins required for efficient *Pf*EMP1 trafficking[12], the specific requirement for PTEX in *Pf*EMP1 export has remained uncertain. Here, we provide evidence of a close interaction between *Pf*EMP1 and PTEX, and corroborate suggestions that PEXEL and PEXEL-negative protein trafficking pathways converge in the PV[10,11,22,41]. In addition, our semiquantitative analysis of *Pf*EMP1B IPs indicated that *Pf*EMP1B is most strongly associated with HSP101. This is consistent with the ability of HSP101 to bind protein cargo separate from PTEX[10] that may occur as early as in the ER[42]. Despite these interactions, we observed only modest immunofluorescence overlap (or PLA signal) between *Pf*EMP1B and HSP101 or EXP2, similar to that observed with endogenous *Pf*EMP1 (ref. 17). This may indicate that *Pf*EMP1, upon entry into the PV, is not relayed directly to the PTEX but instead at least part of the population is maintained in discrete holding or sorting sites before translocation across the PVM. Analysis of the lateral mobility of PV components has suggested the presence of

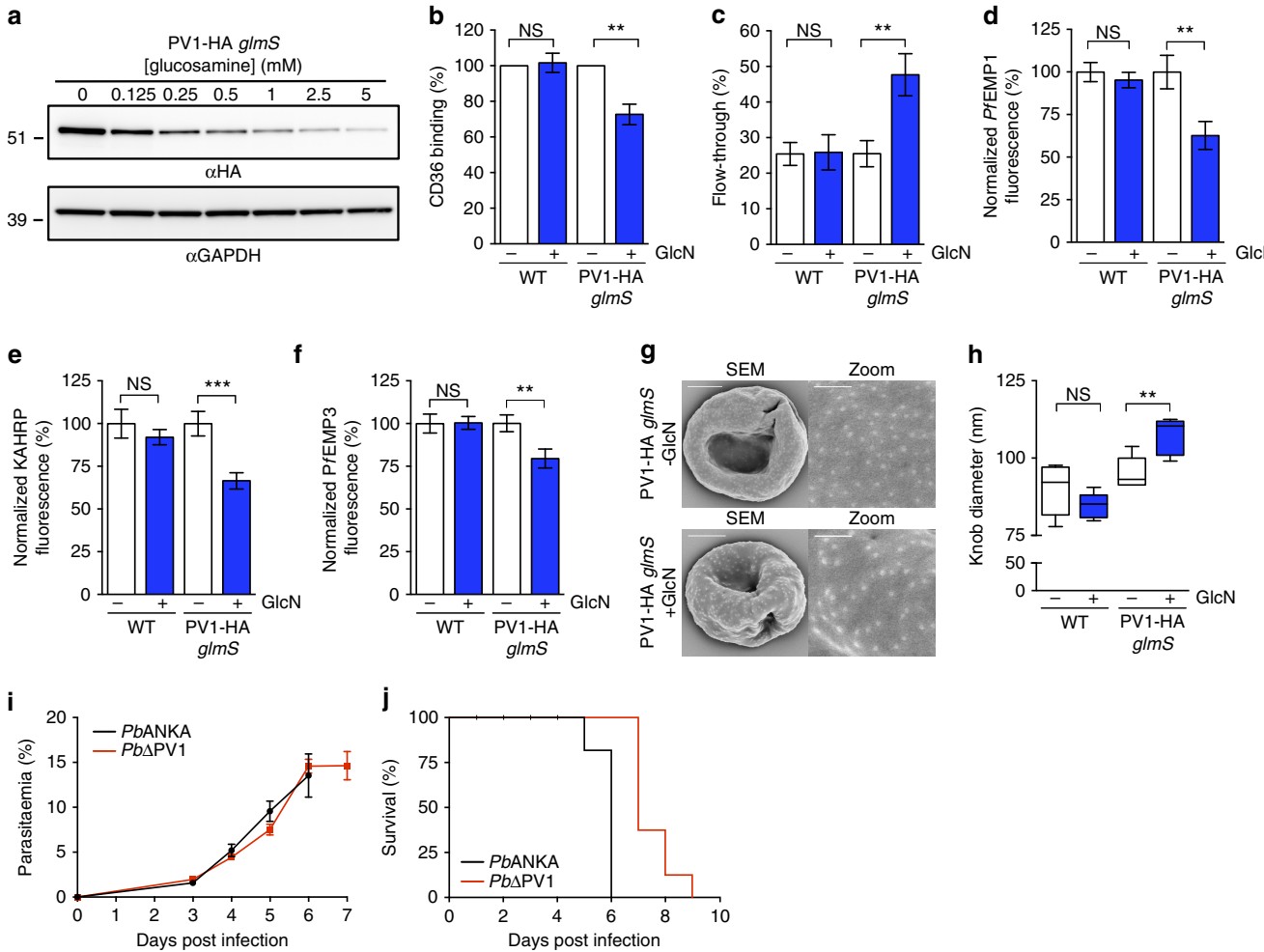

**Figure 7 | Genetic attenuation of PV1 affects virulence determinants in *P. falciparum* and *P. berghei*.** (**a**) PV1-HA *glmS* parasite-infected RBCs treated with increasing concentrations of glucosamine (GlcN) from the trophozoite stage of the previous cycle. Western blots (anti-HA) confirm knockdown of expression of PV1-HA. Molecular masses are shown in kDa. Uncropped western blots are shown in Supplementary Fig. 12. (**b**) Adherence of wild-type (WT) or PV1-HA *glmS*-infected RBCs (23–26 or 30–34 h post invasion (h.p.i.)) to recombinant CD36 under flow conditions (0.1 Pa) ± s.e.m. measured in 10 predetermined fields (binding indicates % of nontreated control, *n* = 5, **P = 0.006 unpaired *t*-test, NS, not significant). (**c**) Passage of WT or PV1-HA *glmS*-infected RBCs (23–26 h.p.i.) through microbeads ± s.e.m. (filterability compares parasitaemia pre- and post-filtration, *n* = 6, **P = 0.01 unpaired *t*-test). (**d–f**) Intensity of *Pf*EMP1, KAHRP and *Pf*EMP3 immunofluorescence signals ± s.e.m. in WT or PV1-HA *glmS*-infected RBCs (23–26 h.p.i.), normalized relative to the signal from the cytoplasmic protein, *Pf*GAPDH (*Pf*EMP1: *n* ≥ 15 cells, **P = 0.0064 unpaired *t*-test; KAHRP: *n* ≥ 56 cells, ***P < 0.001 unpaired *t*-test; *Pf*EMP3: *n* ≥ 26 cells, **P = 0.009 unpaired *t*-test). (**g,h**) Scanning electron microscopy (SEM) of WT or PV1-HA *glmS*-infected RBCs. (**g**) Representative SEM micrographs of knobs from PV1-HA *glmS*-infected RBCs in the presence and absence of 2.5 mM GlcN. Scale bars, 3 μm. Zoom scale bars, 500 nm. (**h**) Quantification of knob size represented by box-and-whisker plot showing median and interquartile range (*n* = 6 cells, **P = 0.003 unpaired *t*-test, whiskers represent minimum/maximum measurements). (**i**) Parasitaemia and (**j**) survival curves of C57/BL6 mice (mean ± s.e.m., *n* = 11 *Pb*ANKA, *n* = 10 *Pb*ΔPV1) after intraperitoneal administration of 1 × 10⁶ *Pb*ANKA or *Pb*ΔPV1 parasites. (**i**) No significant difference in parasitaemia for *Pb*ANKA and *Pb*ΔPV1 parasites was observed (P = 0.63, unpaired *t*-test). (**j**) Survival of *Pb*ΔPV1-infected mice was increased in comparison with *Pb*ANKA-infected mice (P < 0.0001, log-rank (Mantel–Cox) test). Death indicates mice succumbing to cerebral malaria.

distinct subcompartments within the PV[43]. It is possible these compartments provide preferential access to PTEX.

We have identified a parasite protein complex that may help facilitate protein export in these privileged PV compartments. The 1.1 MDa EPIC comprises at least three proteins; a previously identified PTEX-associated protein, PV1 (refs 25,37,44), and two, previously uncharacterized proteins, here termed PV2 and EXP3. Given the molecular masses of these individual proteins, it is likely one or more of these components exist as oligomers, or that other proteins are present in this complex. We demonstrated that EPIC components associate with PTEX, suggesting an interplay between these two complexes. We observed an extensive protein interaction network contiguous with EPIC, and hypothesize that EPIC is involved in the movement of cargo to appropriate PV

subcompartments, representing a nexus for protein sorting in the PV. In the case of exported proteins, this may involve facilitating transfer to PTEX. Indeed, the majority of exported proteins pulled down with EPIC have peak expression in early- to mid-stage trophozoites (when IPs were performed) consistent with active transport of these proteins in the PV. This concept of facilitated sorting in the PV is consistent with the inability of position-5 PEXEL mutants to be exported into the host RBC despite successful entry into the PV[45–48], and provides an explanation for the convergence of the PEXEL and PEXEL-negative trafficking pathways in the PV[10,11,22,41].

We used the *glmS* ribozyme/riboswitch to examine the effect of knockdown of the EPIC components. A previous report indicated that PV1 could not be genetically ablated in *P. falciparum*[44]. We

observed no significant parasite growth retardation upon knockdown of EPIC components *in vitro* in *P. falciparum* or knockout *in vivo* in *P. berghei*. Further work is required to determine the reason for this apparent discrepancy. Mice infected with *Pb*ΔPV1 parasites, however, succumbed to cerebral malaria later than wild-type parasites, indicating PV1 does indeed contribute to parasite virulence. Given the association of EPIC with exported proteins, we probed for deficiencies in protein export. Analysis of fluorescence intensities in age-matched infected RBCs showed a significant decrease in KAHRP, *Pf*EMP1 and *Pf*EMP3 fluorescence at the RBC membrane upon knockdown of *Pf*PV1. Associated with this we observed an increase in cellular deformability with a twofold increase in ability of infected RBCs to pass through a filter designed to mimic splenic fenestrations. In addition, we observed a 29% reduction in the adhesion of infected RBCs to the endothelial ligand CD36 under physiologically relevant conditions. Modifications that stiffen the RBC membrane facilitate adhesion to endothelial cell

ligands by distributing the tensional forces[49]. We propose that the loss of PV1 results in the inefficient export or altered timing of export of *Pf*EMP1 and of proteins that drive remodelling of the host RBC membrane, with a consequent reduction in cellular rigidity and reduced cytoadhesion. The relatively mild phenotype may indicate some level of redundancy in virulence protein export pathways.

It has been suggested that *Plasmodium* spp. make use of human chaperone proteins to aid in the export of parasite proteins[50–52]. We detected a significant complement of human proteins in our *Pf*EMP1B co-immunoprecipitations. Most of these host proteins belonged to the type II chaperonin complex known as TRiC; a large multi-subunit complex involved in folding proteins in an ATP-dependant manner. TRiC has been shown to remain functional in mature RBCs[53] and we suggest that *Pf*EMP1 (and likely other exported cargo) represents a substrate for its activity. TRiC may function in conjunction with structures known as J-dots[52,54]. It has been suggested that

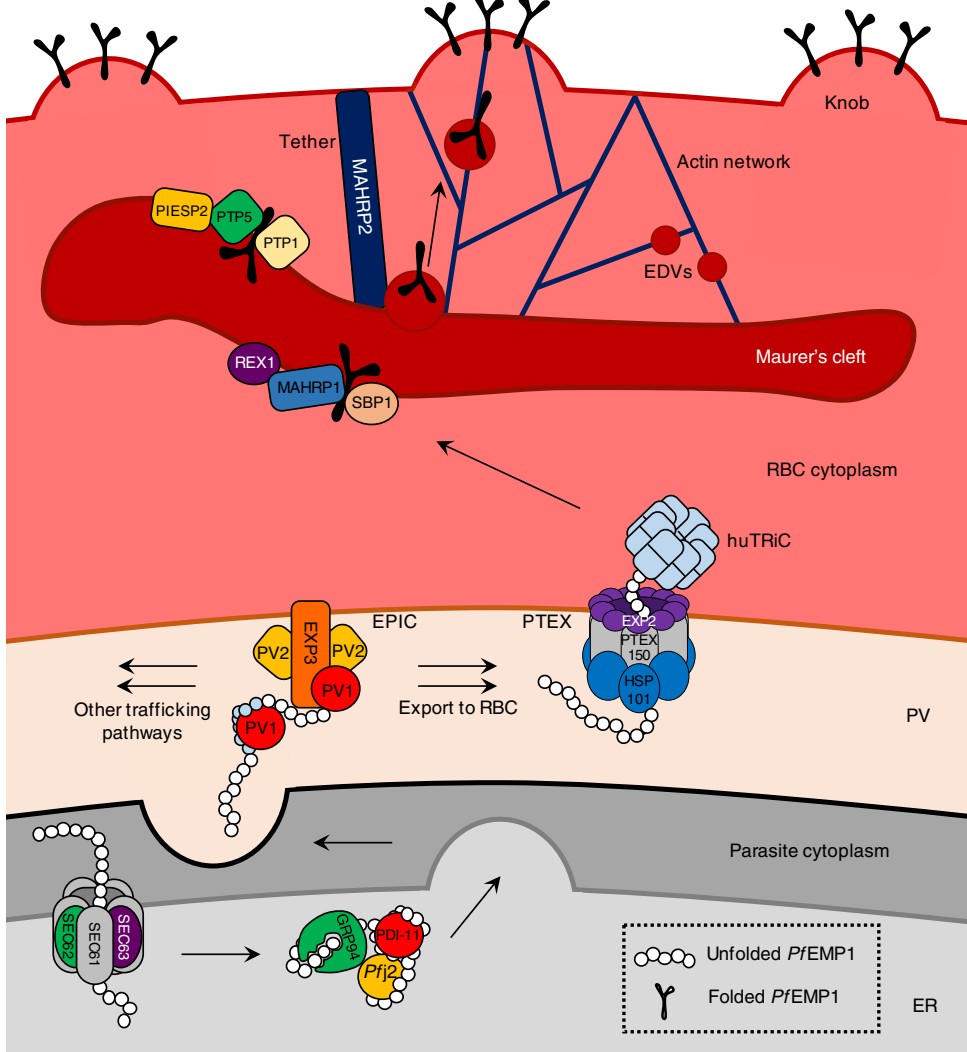

**Figure 8 | Model of *Pf*EMP1 export.** Schematic proposed model of *Pf*EMP1 export mechanisms based on this study. We propose that *Pf*EMP1 is translocated into the ER via a SEC61-62-63-mediated pathway and traffics through the secretory system with the aid of ER-resident chaperones. Upon secretion into the PV, we propose that *Pf*EMP1 is recognized by EPIC and transported to PTEX for export to the RBC. EPIC may also be responsible for the targeting of other cargo to alternate trafficking pathways. After crossing the PVM, we propose that *Pf*EMP1 is received by human TRiC and relayed to the Maurer's clefts, and that Maurer's cleft-resident proteins are involved in stabilizing *Pf*EMP1 at the Maurer's clefts and targeting it to tethers and/or EDVs for export towards the RBC plasma membrane and insertion into knobs. ER, endoplasmic reticulum; PV, parasitophorous vacuole; EPIC, exported protein-interacting complex; PTEX, *Plasmodium* translocon of exported proteins; huTRIC, human TCP1 ring complex; RBC, red blood cell; EDV, electron-dense vesicle.

parasite-encoded TRiC (PfTRiC) is also exported into the host RBC[55]. In agreement with another recent report[56], we were not able to detect any peptides corresponding to PfTRiC components in our co-immunoprecipitations, arguing against a role for PfTRiC in PfEMP1 export.

The Maurer's clefts represent an intermediate compartment in the export of PfEMP1 to the infected RBC surface. We found that SBP1 and REX1, and other Maurer's cleft proteins, interact with PfEMP1 at these structures, consistent with studies showing that the genetic disruption of Maurer's cleft proteins attenuates PfEMP1 surface presentation[23,57–59]. The hydrophobic segment in PfEMP1 is short and has a lower hydropathy index than conventional TM segments[40], and it is possible that Maurer's cleft transmembrane proteins (such as SBP1, MAHRP1 and PTP1 (refs 18,26,30,60)) are needed for docking, insertion and stabilization of a membrane-embedded form of PfEMP1 at the Maurer's clefts. Interestingly, we also observed an interaction of both PfEMP1B and endogenous PfEMP1 with MAHRP2. This is consistent with immuno-ultrastructural evidence showing PfEMP1 association with tethers[14], and points to a potential role for these structures in delivering PfEMP1 to the RBC surface.

In conclusion, we have performed a large-scale protein interaction analysis of PfEMP1. We have identified a number of proteins that interact with PfEMP1 across multiple export compartments from the ER to the PV, to the RBC cytoplasm, and the Maurer's clefts. These data provide significant insights into the pathway for PfEMP1 export. As summarized in the proposed model presented in Fig. 8, we have described a novel complex (EPIC) in the PV that may play a role in delivering proteins in an unfolded state from ER/PV chaperones to PTEX. We show that PV-located PfEMP1 is not membrane embedded and demonstrate an interaction of both PfEMP1 and EPIC with PTEX, consistent with delivery of exported proteins to this transporter. We also show an interaction of PfEMP1 with the host chaperonin complex and propose a mechanism for transfer of PfEMP1 to the Maurer's clefts. We identify Maurer's cleft and tether proteins that interact with PfEMP1 and propose a potential path for delivery of the adhesin to the RBC surface. Moreover, we show that components of this pathway are conserved between P. falciparum and P. berghei and are needed for efficient cytoadherence and virulence. These insights may translate into ways to inhibit the presentation of virulence protein on infected RBCs, and thus ways of interfering with this lethal pathogen.

## Methods

**Parasite culture.** P. falciparum parasites were cultured as previously described[61]. Parasites were cultured in O+ RBC (Australian Red Cross Blood Service) in RPMI with GlutaMAX (Invitrogen) supplemented with 5% human serum (Australian Red Cross Blood Service) and 0.25% AlbuMAX II (Invitrogen). For P. falciparum transfections, ring stage 3D7 (ref. 62) or A4 (ref. 63) strain parasites were electroporated with 100 μg plasmid as previously described[64]. Parasites were maintained on 5 nM WR99210 (Sigma) and cycled in the presence and absence of WR99210 to select for integration into the genome. For P. berghei transfections, knockout vectors were first digested with SacII and XhoI before transfection into wild-type P. berghei ANKA (clone 15cy1)[65]. Pyrimethamine (0.07 mg ml$^{-1}$) was added to the drinking water of mice 1 day post transfection to select for parasites harbouring the knockout vectors. After verification of vector integration, parasite lines were cloned by intravenous injection of a single parasite into the tail vein of BALB/c mice and gene deletion was confirmed by analytical PCR.

**Transgenic parasites.** P. falciparum parasites expressing PfEMP1B, PfEMP1F, EXP1$^{1-35}$-GFP and MAHRP2-GFP were generated previously[13,19,43]. Ribozyme-inducible knockdown constructs were generated by PCR amplification of the last ~1,000 bp of the PfPV1 (PF3D7_1129100), PfPV2 (PF3D7_1226900) and PfEXP3 (PF3D7_1024800) genes. PCR products were ligated into pGlmS-3HA[59] to generate final targeting vectors. P. berghei gene deletions were generated by PCR amplification of ~1.2 kb regions upstream and downstream of the PbPV1, PbPV2 and PbEXP3 coding regions. PCR products were ligated into pBAT[66] to generate final knockout vectors. GFP-tagged exported proteins expressed under the control of the CRT promoter were generated by PCR amplification of full-length

PfMAHRP1 (PF3D7_1370300) and PfPTP5 (PF3D7_1002100) genes, minus the stop codon. PCR products were ligated into pGLUX[67] to generate final expression vectors and transfected into P. falciparum 3D7 parasites. Oligonucleotide sequences used in this study are shown in Supplementary Table 6. The parent lines for transfection were P. falciparum 3D7 (PfPV1, PfMAHRP1 and PfPTP5), A4 (PfPV2 and PfEXP3) and P. berhei ANKA (PbPV1, PbPV2 and PbEXP3).

**Fluorescence microscopy.** Immunofluorescence microscopy was performed on thin blood smears fixed in ice-cold 90% acetone/10% methanol for 5 min. Primary antibodies used included: anti-GFP (mouse, 1:500, Cat. No. 11814460001, Roche), anti-GFP (rabbit, 1:500)[68], anti-V5 (mouse, 1:200, Cat. No. R960-25, Invitrogen), anti-HA (mouse, 1:200, Cat. No. H3663, Sigma), anti-EXP2 (rabbit, 1:500)[9], anti-HSP101 (rabbit, 1:200)[9], anti-MAHRP1 (mouse, 1:500)[18] and anti-PV1 (rabbit, 1:200)[25]. Primary antibodies were detected using Alexa Fluor 488 or 647 secondary antibodies (goat, 1:1,000, Cat. No. A11001, A11008, A21235, A21244, Invitrogen). Parasite nuclei were stained with 2 μg ml$^{-1}$ DAPI (4'-6-diamidino-2-phenylindole). PLA was performed according to the manufacturer's guidelines using the Duolink in situ PLA probes and far-red detection reagent (OLink Bioscience). Microscopy was performed on a DeltaVision Elite restorative widefield deconvolution imaging system (Applied Precision). Live cell imaging of P. berghei was performed using a IX71 microscope (Olympus). Samples were excited using 390, 475 or 632 nm solid-state illumination and imaged using band-pass filters at 435, 523 or 676 nm. Images were taken and deconvoluted using Softworx 5.0 (Applied Precision) and processed using ImageJ software (NIH). Data are presented as average projections of whole-cell Z-stacks with brightness and contrast adjustment. Fluorescence quantification was performed using ImageJ on maximum projection images of whole-cell Z-stacks and the mean fluorescence intensity calculated. For PfEMP1 and KAHRP a 70 × 70 pixel square corresponding to the whole infected RBC was used. For PfEXP3 a 30 × 30 pixel square was chosen to avoid the nonspecific parasite staining. The measurements were normalized to PfGAPDH fluorescence as an internal control. The final values displayed have been adjusted setting the nontreated groups at 100%. The number of individual cells (n) measured is shown in the figure legends.

For subcellular localization immunofluorescence microscopy, labelling of saponin-lysed infected RBCs was performed according to published methods[27]. Infected RBCs were first affixed to slides using Concanavalin A for 15 min before washing three times with phosphate-buffered saline (PBS)[14]. Cells were then lightly fixed in 0.2% formaldehyde added to a solution of 0.015% saponin in PBS for 30 min at 4 °C. After triplicate washing of wells with PBS, primary and secondary antibodies were added as above.

**Protein gel electrophoresis and immunoblotting.** For SDS-PAGE, protein samples were separated on 4–12% Bolt Bis-Tris gels (Invitrogen) and transferred to 0.2 μm nitrocellulose membrane using the iBlot system (Invitrogen). Membranes were blocked in 4% skim milk in PBS overnight at 4 °C. Membranes were probed with the following primary antibodies: anti-GFP (mouse, 1:1,000, Cat. No. 11814460001, Roche), anti-V5 (mouse, 1:500, Cat. No. R960-25, Invitrogen), anti-SBP1 (rabbit, 1:1,000)[23], anti-GAPDH (rabbit, 1:2,000)[24], anti-HSP101 (mouse, 1:100)[9], anti-EXP2 (rabbit, 1:1,000)[9], anti-REX1 (rabbit, 1:1,000)[69], anti-HA (rabbit, 1:500, Sigma), anti-Pf113 (rabbit, 1:1,000)[37], anti-PV1 (rabbit, 1:1,000)[25], anti-Hsp70-2 (mouse, 1:1,000, a kind gift from Gregory Blatch) and anti-Hsp70-x (rabbit, 1:1,000)[54]. Horseradish peroxidase-conjugated secondary antibodies used were anti-mouse (goat, 1:25,000, Cat. No. W4021, Promega) and anti-rabbit (goat, 1:25,000, Cat. No. W4011, Promega). Membranes were incubated with Clarity ECL substrate (Bio-Rad) and imaged using the LAS-3000 imaging system (Fuji). Uncropped immunoblots are presented in Supplementary Fig. 12.

**Native protein analysis.** Mid-trophozoite stage parasites were saponin-lysed and solubilized in 0.5% Triton X-100 for 30 min on ice. The supernatant was diluted in NativePAGE sample buffer (Novex) and Coomassie Brilliant Blue G-250 (0.25% v/v final). Protein samples were separated on 3–12% NativePAGE Bis-Tris gels (Invitrogen) and transferred to 0.2 μm polyvinylidene difluoride membrane using the iBlot system (Invitrogen). Membranes were stained with 0.1% Coomassie R-250 in 50% methanol, 7% acetic acid to mark protein standards and subsequently destained in 50% methanol, 7% acetic acid followed by 90% methanol and 10% acetic acid. Immunoblotting was performed as above.

**Protein solubility analysis.** Late trophozoite stage purified infected RBCs were first subject to hypotonic lysis by resuspension in 10 volumes of ice-cold 2 mM Tris-HCl with complete protease inhibitors (Roche) and incubated for 30 min on ice. The suspension was then clarified at 16,000 × g for 10 min at 4 °C. The supernatant was carefully transferred to a new tube and stored on ice. The pellet was washed three times in PBS and resuspended in 300 μl ice-cold PBS and 100 μl was aliquoted into three tubes. The suspension was again centrifuged at 16,000 × g for 10 min at 4 °C and the supernatant discarded. The resulting pellets were resuspended in 20 volumes of either 0.1 M sodium bicarbonate (Na$_2$CO$_3$, pH 11), 5 M urea or 1% (v/v) Triton X-100 (TX-100) in PBS with complete protease inhibitors (Roche) and incubated for 30 min on ice (or room temperature for 5 M urea extraction). The suspensions were then centrifuged at 16,000 × g for 10 min at

4 °C. The supernatants were carefully transferred to new tubes and stored on ice. The pellets were washed three times in PBS. All soluble and insoluble fractions were subject to SDS-PAGE and Western blot.

**Protease protection assay.** Magnet-purified infected RBCs that were treated with either 0.05% saponin or 3 U of the pore-forming toxin, Tetanolysin, for 10 min at room temperature with shaking at 600 r.p.m. Aliquots were then treated with 20 $\mu$g ml$^{-1}$ proteinase K for 20 min at 37 °C or with PBS. A complete protease inhibitor cocktail (Roche) and phenylmethylsulfonyl fluoride (1 mM) was added for a further 3 min. After centrifugation at 5,000 × g, the pellets were resuspended in reducing sample buffer and analysed by western blotting.

**Immunoprecipitation.** IPs were performed on synchronous mid-trophozoite stage parasites enriched by gelatin flotation or collected with 0.05% saponin in PBS. When stated, enriched trophozoites were first treated with 0.5–1 mM DSP (Pierce) for 30 min. Parasite pellets were solubilized in 10 volumes of IP buffer (1% (v/v) Triton X-100, 150 mM NaCl, 50 mM Tris-HCl, pH 7.4) containing complete protease inhibitors (Roche) for 30 min on ice. Clarified lysates were precleared with 50 $\mu$l protein-A conjugated agarose beads for 30 min and then applied to 25 $\mu$l GFP-Trap (Cat. No. gta-20 Chromotek) or monoclonal anti-HA-agarose (Cat. No. A2905 Sigma) for 4 h or overnight. For IPs using noncommercial matrices, *P. falciparum*-specific antisera were incubated overnight with clarified lysates followed by addition of protein-A agarose beads and a further 2 h of incubation. All IP reactions were washed 5 times with 1 ml IP buffer. For immunoblotting, bound proteins were eluted from beads in 1 × LDS sample buffer. Precleared supernatants were loaded as input lanes (cf. 0.1–2% of eluate lanes) to demonstrate equal parasite material was used in IPs. For mass spectrometry, reactions were further washed twice with 1 mM Tris-HCl pH 7.4 and eluted in 20% (v/v) TFE (2,2,2-Trifluoroethanol) in 0.1% formic acid pH 2.4. After separation of eluate from agarose beads, the pH was neutralized with TEAB (tetraethylammonium bromide) and 5 mM tris(carboxyethyl)phosphine (TCEP) was added. Samples were digested with proteomics grade trypsin (Sigma) overnight.

**Mass spectrometry.** Samples were analysed in triplicate by liquid chromatography (LC) MS/MS using a LTQ Orbitrap Elite (Thermo Scientific) with a nano-electrospray interface coupled to an Ultimate 3,000 RSCL nanosystem (Dionex). The nano-LC system was equipped with an Acclaim Pepmap nano-trap column and an Acclaim Pepmap analytical column. A total of 2 $\mu$l of the peptide mix was loaded onto the trap column at 3% CH3CN containing 0.1% formic acid for 5 min before enrichment column was switched in-line with the analytical column. The LC gradient was 3 to 25% acetonitrile over 20 min, and then 40% acetonitrile over the next 2 min (total run time was 38 min). The LTQ Orbitrap Elite mass spectrometer was operated in the data-dependent mode, spectra acquired first in positive mode at 240 k resolution followed by collision-induced dissociation fragmentation. Twenty of the most intense peptide ions with charge states $\geq 2$ were isolated and fragmented using normalized collision energy of 35 and activation Q of 0.25 (collision-induced dissociation). Mass-spectra (ProteoWizard) were searched against a custom UniProt database containing both the *Homo sapiens* and *P. falciparum* proteomes using MASCOT (Matrix Science). Searches were based on the following criteria for peptide identification: (1) trypsin as digestion enzyme, (2) up to three missed cleavages, (3) 5 p.p.m. peptide tolerance and 0.2 Da fragment tolerance and (4) oxidation of methionine and reduced DSP crosslink on lysine or N terminus as variable modifications. An initial significance threshold of $P < 0.05$ based on a reversed decoy database was used. Peptide searches were reanalysed using the Percolator algorithm to improve the rate of confident peptide identification[70]. Protein identifications were considered valid if $\geq 2$ unique peptide sequences were detected. Statistically significant peptide matches corresponding to specific protein hits in bait IP reactions (or alternatively, enriched fivefold compared with the control) were collated into tables ordered based on fold-change enrichment. Proteins without a signal peptide were excluded from PV1-HA, PV2-HA and EXP3-HA analysis. For charting, the number of peptides displayed represents the average number of peptide matches across all replicates and experiments.

**PfEMP1 cytoadherence assay.** For up-selection of CD36-binding PfEMP1 variants, recombinant human CD36 (125 $\mu$g ml$^{-1}$ in PBS, R&D Systems) was first immobilized on plastic Petri dishes. Gelatin-purified infected RBCs (1% haematocrit) in bicarbonate-free RPMI-HEPES were added and incubated for 1 h at 37 °C. Unbound cells were washed away with bicarbonate-free RPMI-HEPES, and then fresh culture media and RBCs were added and the dish returned to normal culture conditions. Binding assays under constant flow conditions were performed in culture channel chambers (iBidi) coated with recombinant human CD36 (125 $\mu$g ml$^{-1}$ in PBS). Chambers were blocked in 1% bovine serum albumin in PBS for 1 h at 37 °C. Binding assays were performed on synchronized trophozoites (30–34 h.p.i.) that had been treated with glucosamine in the previous cycle. Infected RBCs (1% haematocrit, 3% parasitaemia) were resuspended in bicarbonate-free RPMI and flowed through the chambers at 0.1 Pa using a Harvard Elite 11 syringe pump. Assays were performed at 37 °C and visualized on a DeltaVision Elite restorative widefield deconvolution imaging system (Applied Precision) using

a 60 × objective. Cultures were flowed through the chamber for 5 min and subsequently washed in bicarbonate-free RPMI-HEPES for 10 min under constant flow conditions. The number of bound cells per field was counted for 10 predetermined fields.

**Cellular filtration.** Spleen-mimic filtration assays were performed according to published methods[38,71]. Metal microbeads of two different sizes, 5–15 $\mu$m and 15–25 $\mu$m (96.5% tin, 3% silver and 0.5% copper; Heraeus), were resuspended in 1% AlbuMax II in PBS and layered into an inverted 1 ml filter tip to obtain a bead bed of 5 mm. Filtration was performed on synchronized (23–26 h.p.i.) PV1-HA *glmS*- or 3D7-infected RBCs at ∼5% parasitaemia, 1% haematocrit in 1% AlbuMax II in PBS. Parasites were incubated from 16 h in the previous cycle in the presence or absence of 2.5 mM glucosamine to achieve efficient protein knockdown. The cells were flowed over the bead bed and the percentage of transiting parasites determined by Giemsa analysis of thin smears.

**Scanning electron microscopy.** Collected trophozoite-infected RBCs were washed once in PBS before fixation with 1% glutaraldehyde for 15 min at room temperature. Fixed cells were washed once in water, placed on a glass coverslip, air-dried and sputtered with a 5 nm gold layer as described previously[72]. Backscattered electrons were detected using a Teneo VolumeScope scanning electron microscope (FEI) at 2 kV and 50 pA. Flat cell surfaces were cropped for knob size analysis. The cropped images were subjected to both high-pass and low-pass filters in ImageJ (NIH) to correct brightness gradient and background noises, respectively. Automatic thresholding was applied to segment the electron dense knobs and sizes were assessed using Analyze Particles tool in ImageJ as Feret diameters[73]. Data were collected for >50 knobs from each cell and was curated to remove any features >60 nm or >150 nm to determine the average knob size per cells. The knob sizes of individual cells (n) measured is shown in the figure legends.

**P. falciparum growth assay.** Synchronized trophozoites (30–32 h.p.i.) were seeded into 24-well plates (1% haematocrit, 0.5% parasitaemia) and treated with glucosamine (0 mM or 2.5 mM) in triplicate. Culture media and glucosamine were replenished daily and parasitaemia readjusted to 0.5% on days 2 and 4. Trophozoite parasitaemia was determined by flow cytometry with a FACSCanto II (BD Biosciences) using SYTO 61 red fluorescence nucleic acid stain (Invitrogen). Thin blood smears were taken daily and stained with Giemsa to determine parasite progression through the lifecycle.

**Mouse infection studies.** Female C57/BL6 mice ($n = 6$–11 mice per group; Animal Resources Centre, Western Australia) at 6 weeks of age were infected intraperitoneally with $1 \times 10^6$ *Pb*ΔPV1, *Pb*ΔPV2 or *Pb*ΔEXP3 parasites or *Pb*ANKA wild-type parasites as a control. From 3 days post infection, parasitaemias were monitored daily by Giemsa-stained tail blood smears. To assess parasite virulence, mice were monitored for cerebral malaria symptoms including ataxia and inability to self-right from day 4 post infection[74]. Mice were humanely killed when displaying cerebral malaria symptoms or when the parasitaemia exceeded 20%. All experiments involving the use of animals were performed in strict accordance with the recommendations of the Australian Government and the National Health and Medical Research Council Australian code of practice for the care and use of animals for scientific purposes. The protocols were approved by the Deakin University Animal Welfare Committee (approval number G37/2013 and G16/2014).

**Assessment of cerebral malaria symptoms.** The criteria for scoring cerebral malaria are based on published protocol[74]. There is a progression of symptoms, but mice were scored each time for the following: ruffled fur, hunching, wobbly gait, limb paralysis (including inability to self-right), convulsions and coma. Each symptom was assigned a score of 1. Only when mice scored a cumulative score of $\geq 4$ were they humanely killed. Two researchers independently performed the scoring.

**Statistical analysis.** Statistical analysis and graphing were performed using Prism 6 (GraphPad) and Prism 7 (GraphPad). P values of <0.05 were considered significant.

**Data availability.** The mass spectrometry proteomics data have been deposited to the ProteomeXchange Consortium via the PRIDE[75] partner repository with the data set identifier PXD006155. All other relevant data are available from the authors on request.

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

## Acknowledgements

We thank the Red Cross Blood Service (Melbourne, Australia) for blood products. We thank A/Prof. E. Hanssen (Advanced Microscopy Facility, The University of Melbourne), Dr P. McMillan (Biological Optical Microscopy Platform, The University of Melbourne) for assistance with microscopy and Dr N. Williamson and Dr C.-S. Anh (Mass Spectrometry and Proteomics Facility, The University of Melbourne) for assistance with mass spectrometry. We thank Dr S. Hermann, Dr N. Spillman, Ms S. Kenny and Mr D. Andrews (The University of Melbourne) for technical assistance, Dr J. Przyborski (Philipps University of Marburg) for reagents and Dr J. Smith (Center for Infectious Diseases Research) for reagents and useful discussions. This work was supported by the National Health & Medical Research Council of Australia and the Australia Research Council (to L.T., P.R.G., T.F.de.K.-W. and M.W.A.D.). S.B. was supported by Australian Post Graduate Award. M.W.A.D. was supported by a NHMRC Early Career Fellowship. L.T. is an ARC Laureate Professor.

## Author contributions

S.B., M.W.A.D., P.R.G., T.F.de.K.-W. and L.T. conceived the project, designed the experiments and wrote the paper. S.B., E.M., S.A.C., K.M., B.L., L.D., S.C.C. and M.W.A.D. carried out the experiments. All authors commented on the manuscript.

## Additional information

**Competing interests:** The authors declare no competing financial interests.

