## [Peer Review File · Nature Communications]

Reviewers' comments:

Reviewer #1 (Remarks to the Author):

The manuscript proposed by Batinovic, McHugh, Chisholm, and co-authors describes the protein/protein interactions of the EMP1 protein in *Plasmodium falciparum*-infected red blood cells. The authors have further complement their descriptive interactome with experiments on a specific interactant, namely the parasitophorous vacuolar protein-1, which has a crucial role in this interactive network. This is a conserved trait as shown with *Plasmodium berghei* homologue of PV1. Such insight into the parasite intimate life is of interest. As per the editorial request, I am only commenting on the proteomic analyses included in this report. The fold change threshold of 5 regarding the proteomics comparison is rather conservative and thus, the proteomics results proposed by the authors are rather solid. However, the statistics are not presented and thus not so well supportive.

1. The physical organization of PfEMP1B in the different compartments has been investigated by means of a protease protection assay. While the results of this experiment are convincing, the authors could have established the boundaries of the different domains by monitoring the protein sequence coverage by tandem mass spectrometry.

2. The authors mentioned that they used tandem mass spectrometry for analyzing the PfEMP1B and PfEMP1F pull-downs after SDS-PAGE. They present the list of proteins found enriched with the full construct based on peptide matches (Supplementary Tables 1,2; ≥ 5 -fold enriched, two independent experiments). The material and method section indicates that samples were analysed in triplicate by tandem mass spectrometry. I concluded that the authors have at least six datasets per condition. The authors indicated that "Significant peptide matches from triplicate runs were collated to form the number of peptides per experiment". This is an unusual way to present such data, thus giving the impression that the data are robust for proteins detected in low abundance. In principle, adding three measurements could be a possible alternative rather than performing larger gradients and acquiring more results in the same measurement. However, this questions the experimental set-up and the validation of the proteins with low numbers of spectral counts. Because only a partial view of the proteins detected in this shotgun analysis is presented, I wonder what the real variability of each replicate is. I also wonder how many proteins have been detected per run. Are the statistics robust enough? No specific statistical method is described for assessing the proteomics data nor any statistical data presented while six datasets have been recorded per project, and no assessment of BH-FDR for example is given. For clarity, the authors should submitted their proteomics raw data to a public repository such as PRIDE (EBI) and give more details in their supplementary tables (the twelve results for example per analysis, the corresponding fold change, and associated p value for each protein). The proteins should be ordered based on their fold change (it seems it is the case, but this is not obvious as presented). Some important details are missing such as the gradient length for

the nanoLC.

3. The authors mentioned that "We normalised PTEX peptide counts to the size of the polypeptides". I guess "peptide counts" should be replaced by "MS/MS spectral counts". Such normalization is classical and called NSAF (normalized spectral abundance factor). It allows the authors to rank their identified interactants by their "molarity".

4. The authors noted that 8 components of the human chaperonin complex, the TCP-1 Ring Complex (TRiC) and a known interaction partner of TRiC, HSP90 were detected. Are the authors sure that these proteins are real cellular interactants? I wonder if these proteins are present because of some misfolding of part of the construct used here during the immunoprecipitation (PfEMP1B versus 1F). It is worth to better analyze the variability of these proteins compared to others. The sentence "This suggests that the parasite may utilise host cell chaperonins during PfEMP1 export" should be carefully reevaluated.

5. I have the same concerns regarding the interactome of the EPIC. While the proteomics results proposed by the authors appear rather solid, the merge of the three analytical replicates is rather unusual. Finally, the use of only two biological replicates is not standard in the field and the statistics not given (p values and fold changes for each protein, BH-FDR scores of the whole comparison).

6. The different supplementary Tables are listing "# significant peptide matches". I guess these numbers are "Peptide-to-spectrum matches (PSMs)" or "MS/MS spectral counts" and not peptide counts.

Reviewer #2 (Remarks to the Author):

The authors pull down a pfemp1 mini protein and identify many potential interacting partners by mass spectrometry. As pfemp1 is a key parasite virulence protein this is important and interesting. The authors identify an extensive number of interacting proteins. Many of the interacting proteins appear to be known components of the PTEX complex which is involved in protein export. The authors also identify a novel protein complex that they refer to as the epic complex. Overall this represents a significant amount of data but the significance of most of the interactions and the interactions of the epic complex with other components, remains unclear. Ultimately the authors focus on three proteins PV1,2, exp3. Only knockdown of PV1 seems to result in a phenotype and this is mild - based on this it seems that the epic complex may be involved in export of pfemp1 but is not a key player. Given this it is unclear if the paper is suitable for the journal even if the points below are addressed.

Specific points:

We cannot comment on figure 1d as relevant controls are missing:

Without a control + proteinase K in the absence of EqII or saponin the experiment cannot be interpreted.

Relevant PV proteins should be included to confirm integrity of the PVM etc.

Figure 2: The authors show in figure 1 that the pfemp1b construct colocalises with Exp2 (a component of the PTEX complex). This is convincing. In figure 2 the authors then use proximity ligation to test whether pfemp1b is in proximity of the PTEX complex. This assay does not measure interaction of protein it measures proximity - not association.

Perhaps the authors could elaborate on their interpretation of the data as the PLA signal seems to be in regions of the PV where there is no visible anti-GFP. The reasons for this are unclear.

Figure 2c. The authors should indicate what fraction of total and immuno-precipitated material are loaded.

The authors identify many proteins in the immunoprecipitations of pfemp1b. The interactions appear to be specific but the controls could be more stringent. In considering whether the TricC components interact with pfemp1 it is unclear whether TricC would interact with any exported protein or specifically with pfemp1 as the controls used are not localised in the red cell. The significance of the majority of the interactions is also unclear.

Line 167: authors claim to confirm a direct interaction by doing an IP from a lysate. This does not confirm a direct interaction as other proteins can mediate the interaction. To confirm a direct interaction the authors would need to use purified proteins. This should be re-worded.

Figure 5: authors should present entire gels and not cropped bands. It seems that the bottom half of the gel is presented in the supplement and the top half in the main figure? Both figures seem to be anti HA blots of native gels. They should be presented on a single gel image. If a longer exposure is required to see certain complexes the both exposures should be shown.

line 245 - should this reference be to figure 6?

GlmS experiments:

The authors claim that they can knockdown PV1 2 3 by 90%. Whilst the intensity of the band may decrease by 90% this does not necessarily indicate a 90% reduction in protein level. It would be essential to know what the lower level of detection is to make this claim i.e. load 10% of wild type and show that this can actually be detected.

Figure 7:

In terms of significance this is the key figure but the phenotypes observed are relatively mild in each case.

In analysing cytoadhesion, rigidity, and knob structures, the comparisons should be between wt 3D7 (or parental parasites) +/- GluN and mutant +/- GluN for all experiments and should be presented in a single figure not split between main text and supplement.

7b presents a comparison of cytoadhesion to CD36 in 3D7 and PV1 knockdown cells. This

should be between wt 3D7 (or parental parasites) +/- GluN and mutant +/- GluN. As it is the experiment cannot be interpreted.

This is essential in all experiments but particularly in the cytoadhesion experiments where the authors select for adherent parasites before the experiment i.e. the parasites should be selected then within a few cycles the experiment should be done +/- GluN (the comparison to the parental parasites is relevant but not the key comparison). Authors should not place controls into the supplement - throughout.

In the experiments in which KAHRP fluorescence is quantitated the data is separated between the main text and the supplement. The controls in the supplement seem to have different values to the experiments in the main text. Presumably this was because the controls were not done in the same experiment? Controls should be done at the same time with parasites treated in an identical manner (including growth, fixation, labelling, imaging).

The KAHRP fluorescence is quantitated relative to another exported protein REX1. Presumably this controls to some extent for expression levels. It would seem preferable to use a protein that is not exported as if the PV1 protein is indeed involved in export then export or folding of REX1 may also be affected? The control should be something that will the authors are confident will not change in the mutant i.e. a non-exported protein. If the authors model is correct that KAHRP export is reduced then you would also expect REX1 export to be reduced. Non-exported REX1 may not be labeled as efficiently as exported REX1 - interpretation of this experiment is complicated.

Figure 7f+g. The authors see a relatively small difference in onset of cerebral malaria and no difference in onset of parasitemia in PV1 KO and wt parasites. Was the experiment done blind? As the criteria for cerebral malaria are somewhat subjective did the authors use a quantitative assessment of ataxia and inability to self right? Presumably there is a progression of symptoms during the last days of the experiment?

Figure 8: the model is very speculative - although this OK it could be indicated more clearly that certain aspects are unclear.

Reviewer #3 (Remarks to the Author):

The authors report the discovery of two complexes involved in trafficking of exported proteins in Plasmodium. One complex is in the parasitophorous vacuole, the other in the RBC. The knockout/knockdown phenotypes are not so impressive, but the definition of the complexes is important and very well done.

A number of points need addressing:

In figure 5c, the localization is not perfect. Is that technical? Does it mean something

important?

On line 245, this should be Fig 6, not 5.

Line 253: the reduction at 5 mM is stated but 2.5 mM is the relevant concentration

Lines 256-258: this is not a knockout, so one cannot conclusively state that EPIC components are not essential.

Why normalize to REX1? Couldn't REX1 export be affected? What if you don't normalize?

Line 287: are these means? Fig 7 provides medians.

Measurement of PfEMP1 at the surface is glaringly absent from this report. Please show ATS staining or explain why you cannot.

In Fig 7c, Flow through is the strongest phenotype but it barely reaches significance because of the error. Too bad this isn't bolstered by rigidity measurements.

Fig 7g: Use of one clone in 6 mice is unacceptable. Sometimes parasites get attenuated during cloning in mice. The PV2 and EXP3 parasites serves as extra controls for the "WT" parasites, but at least one more PV1 clone or a complement is necessary.

Supplementary Fig 7: the legend erroneously says PV1

The statement "Molecular weights are shown in kDa" is used repeatedly. MW is massless.

Lines 366-367: the reports are not contrary; this one is a knockdown and the other is a knockout.

There is no discussion of why knockdown of major components of the EPIC complex gives such a mild phenotype.

Line 409: what is meant by important?

NCOMMS-16-24319- Response to Reviewer's comments:

(Reviewer's comments are in italics)

Reviewer #1 (Remarks to the Author):

The manuscript proposed by Batinovic, McHugh, Chisholm, and co-authors describes the protein/protein interactions of the EMP1 protein in Plasmodium falciparum-infected red blood cells. The authors have further complement their descriptive interactome with experiments on a specific interactant, namely the parasitophorous vacuolar protein-1, which has a crucial role in this interactive network. This is a conserved trait as shown with Plasmodium berghei homologue of PV1. Such insight into the parasite intimate life is of interest. As per the editorial request, I am only commenting on the proteomic analyses included in this report. The fold change threshold of 5 regarding the proteomics comparison is rather conservative and thus, the proteomics results proposed by the authors are rather solid. However, the statistics are not presented and thus not so well supportive.

1. The physical organization of PfEMP1B in the different compartments has been investigated by means of a protease protection assay. While the results of this experiment are convincing, the authors could have established the boundaries of the different domains by monitoring the protein sequence coverage by tandem mass spectrometry.

This suggestion presumably refers to the truncated fragment which was detected at a size of 90-95 kDa using V5 antisera (Fig. 1d; second lane). We noted that this fragment is consistent with the N-terminal (GFP tagged) domain (and TM domain) of Maurer's clefts membrane-embedded PfEMP1. That is, cleavage of the cytoplasmically-exposed C-terminal domain is predicted to give a band of the observed size. Our result for PfEMP1B confirms a previous report for endogenous PfEMP1¹. It is also consistent with many other protease protection studies in which membrane-embedded PfEMP1 has been protease digested from either side of the membrane. Therefore we believe it is reasonable to assume the identity of this fragment without further confirmation.

2. The authors mentioned that they used tandem mass spectrometry for analyzing the PfEMP1B and PfEMP1F pull-downs after SDS-PAGE. They present the list of proteins found enriched with the full construct based on peptide matches (Supplementary Tables 1,2; ≥ 5-fold enriched, two independent experiments). The material and method section indicates that samples were analysed in triplicate by tandem mass spectrometry. I concluded that the authors have at least six datasets per condition. The authors indicated that "Significant peptide matches from triplicate runs were collated to form the number of peptides per experiment". This is an unusual way to present such data, thus giving the impression that the data are robust for proteins detected in low abundance. In principle, adding three measurements could be a possible alternative rather than performing larger gradients and acquiring more results in the same measurement. However, this questions the experimental set-up and the validation of the proteins with low numbers of spectral counts. Because only a partial view of the proteins detected in this shotgun analysis is presented, I wonder what the real variability of each replicate is. I also wonder how many proteins have been detected per run. Are the statistics robust enough? No specific statistical method is described for assessing the proteomics data nor any statistical data presented while six datasets have been recorded per project, and no assessment of BH-FDR for example is given. For clarity, the authors should submitted their proteomics raw data to a public repository such as PRIDE (EBI) and give more details in their supplementary tables (the twelve results for example per analysis, the corresponding fold change, and associated p value for each protein). The proteins should be ordered based on their fold change (it seems it is the case, but this is not obvious as presented). Some important details are missing such as the gradient length for the nanoLC.

The reviewer's suggestion have been followed – as detailed below:

The reviewer is correct that there are six datasets per condition (technical triplicates and two independent experiments).

We have now performed bioinformatics analysis on each replicate individually and presented this data in Supplementary information.

We have analysed the first two replicates of each experiment (four datasets per condition) and have uploaded all six data sets to PRIDE for public viewing.

We note that inclusion of technical duplicates is more rigorous than is usual for published mass spectrometry data sets.

As a result of reanalysis of each replicate individually, the number of interacting proteins identified confidently (at least 2 unique peptide sequences) is slightly reduced.

- For the *PfEMP1B* experiments this results in a reduction of 2 proteins from the total (68 -> 66 proteins). For the EPIC (PV1, PV2 and EXP3) experiments, we observed a combined reduction of 5 proteins (71 -> 66 proteins) in total. These changes are reflected in both the main text and figures, as well as the supplementary tables.
- The removal of these few proteins does not affect the conclusions from the study as the proteins that were analysed and/or discussed further were at the top of the lists. A number of these were validated by Western blotting, where reagents were available.

We have now revised the presentation of the data as follows:

Proteins are now ordered by fold-change and the legend has been changed to state that “proteins are ordered based on their fold change”.

- We note that this may cause some protein hits to appear they are under our ≤ 5 -fold enriched threshold. Given these particular hits are unique to the bait IP (not present in the control reactions), we consider these equally validated hits. To prevent confusion we have amended the supplementary table legends to indicate ‘Parasite proteins that are **unique to, or ≥ 5 -fold** enriched in, the *PfEMP1B* co-IP compared to control *PfEMP1F*’. The table legends representing the other mass spectrometry experiments were amended in line with this example.

The number of proteins detected per run (at least 1 significant peptide) is shown in the table below. This information can also be accessed via the proteomics repository.

IP	Experiment 1 Replicate 1	Experiment 1 Replicate 2	Experiment 2 Replicate 1	Experiment 2 Replicate 2
PfEMP1B	173	150	217	235
PfEMP1F	206	247	300	300
PV1	282	285	274	269
3D7 (PV1 control)	119	132	70	129
PV2	112	107	207	191
EXP3	101	89	165	181
A4 (PV2/EXP3 control)	103	99	155	99

The statistical methods for assessing the proteomics data and statistical data are now provided in the methods section.

The assessment of BH-FDR is provided. The FDR value is included at the bottom of the new supplementary tables. There is no protein level p value, but individual peptides have expected scores. These data can be accessed via the proteomics repository.

The proteomics raw data has been submitted to a public repository (PRIDE; EBI) under the accession number PXD006155 and will be available publicly once the manuscript is published. This data is privately available to reviewers using the following login details:

Username: reviewer84384@ebi.ac.uk

Password: EtPjZi0x

We have now provided more details in the supplementary tables. These include the 4 results per analysis, the fold-change and FDR, and the expected scores for individual peptides. These data can be accessed via the proteomics repository.

The mass spectrometry methods section has been expanded to include additional information as to how experiments were performed, including the gradient length for the nanoLC.

3. The authors mentioned that “We normalised PTEX peptide counts to the size of the polypeptides”. I guess “peptide counts” should be replaced by “MS/MS spectral counts”. Such normalization is classical and called NSAF (normalized spectral abundance factor). It allows the authors to rank their identified interactants by their “molarity”.

The term “peptide counts” has been replaced by “MS/MS spectral counts”.

4. The authors noted that 8 components of the human chaperonin complex, the TCP-1 Ring Complex (TRiC) and a known interaction partner of TRiC, HSP90 were detected. Are the authors sure that these proteins are real cellular interactants? I wonder if these proteins are present because of some misfolding of part of the construct used here during the immunoprecipitation (PfEMP1B versus 1F). It is worth to better analyze the variability of these proteins compared to others. The sentence “This suggests that the parasite may utilise host cell chaperonins during PfEMP1 export” should be carefully reevaluated.

In the submitted version of the m/s we provided pull-downs with PfEMP1F as a control and showed that RBC TRiC was not pulled-down, indicating that non-specific interactions do not occur during the pull-down procedure. However the reviewer makes the important point that RBC TRiC and Hsp90 may interact with a number of exported proteins that need help in folding after passage through the PTEX. To examine this possibility we undertook pull-down experiments with another exported protein, [redacted]. This suggests that the interactions may be quite promiscuous. We have not pursued this further and have not included these data in the revised manuscript, as this is not a major point for this manuscript. Instead, we have modified the text of the manuscript to remove any suggestion that RBC chaperones interact only with PfEMP1 and state that they may interact with a range of exported proteins that require help in folding. We are however willing to include this data in the Suppl Info if the Editor thinks it is useful.

5. I have the same concerns regarding the interactome of the EPIC. While the proteomics results proposed by the authors appear rather solid, the merge of the three analytical replicates is rather unusual. Finally, the use of only two biological replicates is not standard in the field and the statistics not given (p values and fold changes for each protein, BH-FDR scores of the whole comparison).

The approach we have adopted is similar to other recent publication in the malaria field. See for example: Koning Ward et al 2009², Josling et al 2015,³ Mesén-Ramírez et al 2016⁴. Moreover, a number of the interaction partners in our manuscript were validated by Western blotting.

6. The different supplementary Tables are listing “# significant peptide matches”. I guess these numbers are “Peptide-to-spectrum matches (PSMs)” or “MS/MS spectral counts” and not peptide counts.

As mentioned above, the term “# significant peptide matches” has been changed to “Peptide-to-spectrum matches (PSMs)” or “MS/MS spectral counts” in the supplementary Tables.

Reviewer #2 (Remarks to the Author):

The authors pull down a pfemp1 mini protein and identify many potential interacting partners by mass

spectrometry. As *pfemp1* is a key parasite virulence protein this is important and interesting. The authors identify an extensive number of interacting proteins. Many of the interacting proteins appear to be known components of the PTEX complex which is involved in protein export. The authors also identify a novel protein complex that they refer to as the epic complex. Overall this represents a significant amount of data but the significance of most of the interactions and the interactions of the epic complex with other components, remains unclear. Ultimately the authors focus on three proteins PV1,2, *exp3*. Only knockdown of PV1 seems to result in a phenotype and this is mild - based on this it seems that the epic complex may be involved in export of *pfemp1* but is not a key player. Given this it is unclear if the paper is suitable for the journal even if the points below are addressed.

Specific points:

We cannot comment on figure 1d as relevant controls are missing: Without a control + proteinase K in the absence of EqII or saponin the experiment cannot be interpreted. Relevant PV proteins should be included to confirm integrity of the PVM etc.

The Equinatoxin-II reagent which we used for the previous experiments is no longer available. In order to provide the additional controls requested by the reviewer we have repeated this experiment, using an alternative pore-forming toxin, tetanolysin. The new data are provided as Figure 1c,d. This experiment additionally contains a PV-localised protein control, PV1. The tetanolysin experiment leads to the same conclusion as for the Equinatoxin experiment. The use of two different RBC permeabilizing reagents strengthens our conclusions.

The Reviewer also asked for a control in which intact infected RBCs were treated to proteinase K in the absence of saponin or the pore-forming toxin. This experiment is not directly related to the data presented in Fig 1c,d, and is technically difficult because the host cell contents (from intact infected RBCs) complicates the Western analysis. Nonetheless, at the reviewer's request, we have subjected intact infected RBCs to proteinase K treatment. We have provided the data at the end of this response letter (Fig A). In agreement with previous reports for PfEMP1b^{5 6} (and for endogenous PfEMP1⁷), which concluded that only a minor population of PfEMP1 is surface exposed, we observed no detectable digestion of PfEMP1 when proteinase K is applied to intact cells. This experiment has technical limitations due to the relatively low amount of intact infected RBCs that can be applied to a gel which may prevent detection of the small population of surface-exposed PfEMP1b. However the data confirm that most of the PfEMP1 is located in internal compartments and therefore is not accessible to proteinase K at the RBC surface. This validates our interpretation of the data in Figure 1. We have not included the data in the manuscript given the technical limitations of this experiment and the fact that it does not fit well into Figure 1 or Suppl Figure 1. (These experiments having all been performed using permeabilised cells). However we are willing to include the data as part of the Supplementary material if the Editor or reviewer feels that this is required.

*Figure 2: The authors show in figure 1 that the *pfemp1b* construct colocalises with *Exp2* (a component of the PTEX complex). This is convincing. In figure 2 the authors then use proximity ligation to test whether *pfemp1b* is in proximity of the PTEX complex. This assay does not measure interaction of protein is measures proximity - not association.*

We accept this point and have changed the text accordingly.

Perhaps the authors could elaborate on their interpretation of the data as the PLA signal seems to be in regions of the PV where there is no visible anti-GFP. The reasons for this are unclear.

The reviewer is correct that the PLA signal is strongest in some regions that have relatively little GFP signal. This indicates that only a sub-population of PfEMP1 interacts with the PTEX at steady state. We have now

emphasised this in the Results section and noted in the Discussion that: “This may indicate that PfEMP1, upon entry into the PV, is not relayed directly to the PTEX but instead at least part of the population is maintained in discrete holding or sorting sites prior to translocation across the PVM.”

Figure 2c. The authors should indicate what fraction of total and immuno-precipitated material are loaded.

Figure 2c. We have now included information about the fraction of total and immuno-precipitated material that were loaded. Pre-cleared supernatants were loaded as input lanes (cf. 0.1-2% of eluate lanes) to demonstrate equal parasite material was used in immunoprecipitations experiments.

The authors identify many proteins in the immunoprecipitations of pfemp1b. The interactions appear to be specific but the controls could be more stringent. In considering whether the TricC components interact with pfemp1 it is unclear whether TricC would interact with any exported protein or specifically with pfemp1 as the controls used are not localised in the red cell. The significance of the majority of the interactions is also unclear.

As described above, we now have evidence that RBC TRiC may interact in a promiscuous manner with exported proteins, and may help in folding different exported proteins after passage through the PTEX. We have modified the text accordingly.

Line 167: authors claim to confirm a direct interaction by doing an IP from a lysate. This does not confirm a direct interaction as other proteins can mediate the interaction. To confirm a direct interaction the authors would need to use purified proteins. This should be re-worded.

As noted in the text there is evidence in the literature that is consistent with an interaction of PfEMP1 with several of the top identified hits: namely SBP1, REX1, MAHRP1, MAHRP2, PTP5 and the PTEX complex. We have followed up on the potential functional significance of three new proteins and provide evidence for a role for PV1. We believe that our work opens new avenues for the exploration of a number of new proteins involved in PfEMP1 trafficking.

Line 167: We accept the reviewer’s point that IP from a lysate does not confirm a direct interaction. We have re-worded the relevant sentences to indicate that these interactions may be direct or indirect.

Figure 5: authors should present entire gels and not cropped bands. It seems that the bottom half of the gel is presented in the supplement and the top half in the main figure? Both figures seem to be anti HA blots of native gels. They should be presented on a single gel image. If a longer exposure is required to see certain complexes the both exposures should be shown.

We assume the reviewer is referring to Fig 5a which is a blot from a native gel. We split the gels into two parts to illustrate different points and to save space in the main figure section. We have now included the full-length gels as part of Suppl Fig 6, so that both the smaller and larger complexes can be observed together.

line 245 - should this reference be to figure 6?

Line 245. We apologise. This statement does refer to figure 6. This has now been corrected.

GlmS experiments:

The authors claim that they can knockdown PV1 2 3 by 90%. Whilst the intensity of the band may decrease by 90% this does not necessarily indicate a 90% reduction in protein level. It would be essential to know what the lower level of detection is to make this claim i.e. load 10% of wild type and show that this can actually be detected.

This has now been performed. The result is shown in Fig B at the bottom of this response. The limit of detection is reached at ~3% of the loaded sample. This suggests that our estimates of the level of knockdown may be slightly overestimated. To indicate this we have changed the text to refer to the “*apparent level of knockdown*”.

Figure 7: In terms of significance this is the key figure but the phenotypes observed are relatively mild in each case. In analysing cytoadhesion, rigidity, and knob structures, the comparisons should be between wt 3D7 (or parental parasites) +/- GluN and mutant +/- GluN for all experiments and should be presented in a single figure not split between main text and supplement.

7b presents a comparison of cytoadhesion to CD36 in 3D7 and PV1 knockdown cells. This should be between wt 3D7 (or parental parasites) +/- GluN and mutant +/- GluN. As it is the experiment cannot be interpreted.

This is essential in all experiments but particularly in the cytoadhesion experiments were the authors select for adherent parasites before the experiment i.e. the parasites should be selected then within a few cycles the experiment should be done +/- GluN (the comparison to the parental parasites is relevant but not the key comparison).

Authors should not place controls into the supplement - throughout.

In the experiments in which KAHRP fluorescence is quantitated the data is separated between the main text and the supplement. The controls in the supplement seem to have different values to the experiments in the main text. Presumably this was because the controls were not done in the same experiment? Controls should be done at the same time with parasites treated in an identical manner (including growth, fixation, labelling, imaging).

We have now included all of the relevant data into Figure 7 so that side-by-side comparisons can be made.

The KAHRP fluorescence is quantitated relative to another exported protein REX1. Presumably this controls to some extent for expression levels. It would seem preferable to use a protein that is not exported as if the PV1 protein is indeed involved in export then export or folding of REX1 may also be affected? The control should be something that will the authors are confident will not change in the mutant i.e. a non-exported protein. If the authors model is correct that KAHRP export is reduced then you would also expect REX1 export to be reduced. Non-exported REX1 may not be labeled as efficiently as exported REX1 - interpretation of this experiment is complicated.

These experiments have been repeated and the immunofluorescence signals for exported proteins (KAHRP, PfEMP3, PfEMP1) have now been quantitated relative to PfGAPDH (Fig 7d).

Figure 7f+g. The authors see a relatively small difference in onset of cerebral malaria and no difference in onset of parasitemia in PV1 KO and wt parasites. Was the experiment done blind? As the criteria for cerebral malaria are somewhat subjective did the authors use a quantitative assessment of ataxia and inability to self right? Presumably there is a progression of symptoms during the last days of the experiment?

Additional information about the criteria (which are based on published protocols⁸) are now included in Supplementary Methods. There is a progression of symptoms, but mice were scored each time for the following: ruffled fur, hunching, wobbly gait, limb paralysis (including inability to self-right), convulsions and coma. Each symptom was assigned a score of 1. Only when mice scored a cumulative score of ≥ 4 were they humanely culled. Two researchers independently performed the scoring.

Figure 8: the model is very speculative - although this OK it could be indicated more clearly that certain aspects are unclear.

This has been now been made clear.

Reviewer #3 (Remarks to the Author):

The authors report the discovery of two complexes involved in trafficking of exported proteins in Plasmodium. One complex is in the parasitophorous vacuole, the other in the RBC. The knockout/knockdown phenotypes are not so impressive, but the definition of the complexes is important and very well done.

A number of points need addressing:

In figure 5c, the localization is not perfect. Is that technical? Does it mean something important?

This is consistent with our model whereby EPIC holds cargo in the PV lumen before transferring that cargo to the PTEX. Thus incomplete overlap of the fluorescence signals might be expected. We have now included a sentence in the text that explains that “This may indicate that PfEMP1, upon entry into the PV, is not relayed directly to the PTEX but instead at least part of the population is maintained in discrete holding or sorting sites prior to translocation across the PVM.”

On line 245, this should be Fig 6, not 5

We apologise. This statement does refer to figure 6. This has now been corrected.

Line 253: the reduction at 5 mM is stated but 2.5 mM is the relevant concentration.

We have now revised the text to report the apparent reduction in signal at 2.5 mM and presented the densitometric analyses in Sup. Figure 7b.

Lines 256-258: this is not a knockout, so one cannot conclusively state that EPIC components are not essential.

We accept the reviewer’s comment that we cannot conclusively state that EPIC components are not essential in *P. falciparum* given that we have not generated a knockout. We have changed the text accordingly.

Why normalize to REX1? Couldn't REX1 export be affected? What if you don't normalize?

As indicated above in the response to Reviewer 2, we have now normalized the fluorescence signal to the parasite cytoplasmic protein, PfGAPDH.

Line 287: are these means? Fig 7 provides medians.

In the submitted manuscript we provided immunofluorescence quantification as bars and whiskers plots and the median values were indicated. In the revised manuscript, (at the reviewer’s request) we have normalised the immunofluorescence signal to the mean fluorescence value of the cytoplasmic protein, PfGAPDH. To avoid confusion we have now instead presented the data as bar graphs and indicated the mean +/- S.E.M. We have retained the bars and whiskers plots for the knob diameter data and have indicated in the figure legend that the median value is presented.

Measurement of PfEMP1 at the surface is glaringly absent from this report. Please show ATS staining or explain why you cannot.

We have now quantified the level of ATS labelling (see Fig 7d). This antibody strongly recognises PfEMP1 at the Maurer's clefts but does not recognise surface-embedded PfEMP1 (likely because most of the PfEMP1 is trapped in intracellular compartments, plus the epitope may be obscured in the small surface-embedded fraction). We saw a similar decrease in the PfEMP1 signal in PV1 knock-down parasites, indicating that PV1 plays a role in trafficking of PfEMP1 to the Maurer's clefts. The text has been modified accordingly.

In Fig 7c, Flow through is the strongest phenotype but it barely reaches significance because of the error. Too bad this isn't bolstered by rigidity measurements.

We have now performed an additional experiment which has resulted in an increase in the significance of the observed difference (Fig 7 c).

Fig 7g: Use of one clone in 6 mice is unacceptable. Sometimes parasites get attenuated during cloning in mice. The PV2 and EXP3 parasites serves as extra controls for the "WT" parasites, but at least one more PV1 clone or a complement is necessary.

We have now performed an additional experiment with mice infected with a separate clone of *Pb*ΔPV1 parasites compared with mice infected with wildtype *Pb*ANKA (Supplementary Fig. 11c).

Supplementary Fig 7: the legend erroneously says PV1.

PV1 is mentioned in the legend because, while the figure presents quantitation data for PV1, PV2 and Exp3, as well as Western data for PV2 and Exp3.

The statement "Molecular weights are shown in kDa" is used repeatedly. MW is massless

The statement "Molecular weights are shown in kDa" has been changed to "Molecular masses are shown in kDa".

Lines 366-367: the reports are not contrary; this one is a knockdown and the other is a knockout.

The text has been changed to remove reference to the reports being contrary and to note that further work is required to determine the reason for the apparent differences in the two studies.

There is no discussion of why knockdown of major components of the EPIC complex gives such a mild phenotype.

We have now included a statement that says: "The relatively mild phenotype in the *P. falciparum* may indicate some level of redundancy in virulence protein export pathways."

Line 409: what is meant by important?

Line 409: The word "important" has been removed.

References

1. Kriek N, Tilley L, Horrocks P, et al. Characterization of the pathway for transport of the cytoadherence-mediating protein, PfEMP1, to the host cell surface in malaria parasite-infected erythrocytes. *Mol Microbiol.* 2003;50(4):1215-1227.
2. de Koning-Ward TF, Gilson PR, Boddey JA, et al. A newly discovered protein export machine in malaria parasites. *Nature.* 2009;459(7249):945-949.

3. Josling GA, Petter M, Oehring SC, et al. A *Plasmodium falciparum* bromodomain protein regulates invasion gene expression. *Cell Host Microbe*. 2015;17(6):741-751.
 4. Mesen-Ramirez P, Reinsch F, Blancke Soares A, et al. Stable translocation intermediates jam global protein export in *Plasmodium falciparum* parasites and link the PTEX component EXP2 with translocation activity. *PLoS Pathog*. 2016;12(5):e1005618.
 5. Melcher M, Muhle RA, Henrich PP, et al. Identification of a role for the PfEMP1 semi-conserved head structure in protein trafficking to the surface of *Plasmodium falciparum* infected red blood cells. *Cell Microbiol*. 2010;12(10):1446-1462.
 6. McMillan PJ, Millet C, Batinovic S, et al. Spatial and temporal mapping of the PfEMP1 export pathway in *Plasmodium falciparum*. *Cell Microbiol*. 2013;15(8):1401-1418.
 7. Waterkeyn JG, Wickham ME, Davern KM, et al. Targeted mutagenesis of *Plasmodium falciparum* erythrocyte membrane protein 3 (PfEMP3) disrupts cytoadherence of malaria-infected red blood cells. *EMBO J*. 2000;19(12):2813-2823.
 8. de Oca MM, Engwerda C, Haque A. *Plasmodium berghei* ANKA (PbA) infection of C57BL/6J mice: a model of severe malaria. *Methods Mol Biol*. 2013;1031:203-213.
-

Figure A. Treatment of intact infected RBCs with proteinase K. Late stage PfEMP1B-GFP transfectants were Percoll-purified and incubated in PBS plus (+) or minus (-) proteinase K. Bands of the expected size for PfEMP1B are observed when the blot is probed with anti-ATS and anti-GFP antibodies. Little to no degradation is observed in the proteinase K treated sample. Equal loading is shown by probing with anti-ERC antibodies.

Method

Late stage parasites were Percoll-purified. Purified parasites were washed in 1X PBS and incubated with or without proteinase K (final 1mg/mL) for 30 minutes on ice. The parasite samples were TCA precipitated, washed in acetone twice, and air dried. The samples were resuspended in Bolt sample buffer (Invitrogen) and subjected to electrophoresis on a 3-8% Tris acetate gel, then transferred to a nitrocellulose membrane for Western blotting. The following primary antibodies were used. Anti-ATS (PfEMP1 MAb 1B/98-6H1-1, 1:100), anti-GFP (Roche MAb 1:500) and anti-ERC (1:1000, rabbit). Anti-mouse/rabbit-HRP conjugated secondary antibodies were used (1:15000, Millipore). Western blotting was performed as described in the Methods section.

Figure B. Detection limit of anti-HA antibodies by Western blot. Trophozoite stage PV1-HA *glmS* transfectants were loaded in decreasing amounts. A band of the expected size was observed when the blot is probed with anti-GFP antibodies. The detection limit is at ~3% of the loaded sample.

REVIEWERS' COMMENTS:

Reviewer #1 (Remarks to the Author):

Batinovic and co-authors investigated the protein/protein interactions of the EMP1 protein in Plasmodium falciparum-infected red blood cells. The authors answered my previous concerns and appropriately amended their interesting manuscript. The supplementary data and proteomics data deposited on the PRIDE repository are well supporting their results. I am convinced that this piece of work is helpful for further investigations on the EMP1 interactome.

Reviewer #2 (Remarks to the Author):

Overall the quality of the data has improved. As in the previous report it is clear that the components identified to not play a major role in the export of pFEMP1. The effects seen in most functional assays are relatively small (this is the key part of the paper). This is not the first report identifying proteins involved in export of pfemp1. Whilst the data is interesting it is up to the journal to decide whether this reaches the level of novelty required for publication - had functional phenotypes been more pronounced it likely would.

Reviewer #3 (Remarks to the Author):

The authors have nicely addressed the reviewers comments. The paper is well done and makes a strong contribution to the field.

Reviewer #1 (Remarks to the Author):

Batinovic and co-authors investigated the protein/protein interactions of the EMP1 protein in *Plasmodium falciparum*-infected red blood cells. The authors answered my previous concerns and appropriately amended their interesting manuscript. The supplementary data and proteomics data deposited on the PRIDE repository are well supporting their results. I am convinced that this piece of work is helpful for further investigations on the EMP1 interactome.

We are pleased that we have addressed the Reviewer's comments

Reviewer #2 (Remarks to the Author):

Overall the quality of the data has improved. As in the previous report it is clear that the components identified to not play a major role in the export of pFEMP1. The effects seen in most functional assays are relatively small (this is the key part of the paper). This is not the first report identifying proteins involved in export of pfemp1. Whilst the data is interesting it is up to the journal to decide whether this reaches the level of novelty required for publication - had functional phenotypes been more pronounced it likely would.

We have identified a novel complex (EPIC) involved in virulence protein trafficking. Further, we showed that knock-down of an EPIC component results in a ~40% decrease in PfEMP1 export, a ~30% decrease in the ability of infected RBCs to cytoadhere to endothelial cell ligands, and a ~50% decrease in the rigidity of the infected RBC membrane. Moreover, we show that deletion of the *P. berghei* homologue of PV1 is associated with substantive attenuation of parasite virulence *in vivo*. We accept the reviewer's point that the loss of function is not complete; however one would anticipate that there would be some level of redundancy in a pathway that is as important as the virulence protein export pathway. While previous work has identified other proteins involved in trafficking and presentation of PfEMP1, these have all been located at the Maurer's clefts or the RBC membrane. The EPIC complex appears to represent novel trafficking machinery located in the parasitophorous vacuole. We believe our work provides important new insights into virulence mechanisms in plasmodium.

Reviewer #3 (Remarks to the Author):

The authors have nicely addressed the reviewers comments. The paper is well done and makes a strong contribution to the field.

We are pleased that we have addressed the Reviewer's comments and note that the reviewer agrees that we make a strong contribution to the field.